# Collapse or Thrive?
# Perils and Promises of Synthetic Data in a Self-Generating World

Joshua Kazdan [* 1]   Rylan Schaeffer [* 2]   Apratim Dey [1]   Matthias Gerstgrasser [2]   Rafael Rafailov [2]
David Donoho [1]   Sanmi Koyejo [2]

## Abstract

What happens when generative machine learning models are pretrained on web-scale datasets containing data generated by earlier models? Some prior work warns of "model collapse" as the web is overwhelmed by synthetic data; other work suggests the problem can be contained by managing how available data are used in pretraining. We report experiments on three ways of using data (training-workflows), across three generative model task-settings (multivariate Gaussian estimation, kernel density estimation, and language-model fine-tuning) to further confirm the possibility of containment: (a) we confirm that the training-workflow of *replacing* all real data by successive generations of purely synthetic data suffers model collapse; (b) we consider the training-workflow of *accumulating* synthetic data alongside real data and training on all data combined and confirm that, although the proportion of real data eventually becomes zero, models remain stable and their test losses do not diverge under this training-workflow; (c) we consider a training-workflow where real and synthetic data accumulate together but successive generations of pretraining are constrained to use fixed-size data subsets each generation. In this workflow, we observe slow and gradual rather than explosive degradation of test loss performance across generations. Our insights are important when forecasting whether future generative models will collapse or thrive, and our results open avenues for empirically and mathematically studying the context-dependent value of synthetic data.

*Equal contribution [1]Stanford Statistics [2]Stanford Computer Science. Correspondence to: Joshua Kazdan <jkazdan@stanford.edu>, Rylan Schaeffer <rschaef@cs.stanford.edu>, Sanmi Koyejo <sanmi@cs.stanford.edu>.

*Proceedings of the $42^{dn}$ International Conference on Machine Learning*, Vancouver, Canada. PMLR 267, 2025. Copyright 2025 by the author(s).

## 1. Introduction

With each passing day, the internet contains more AI-generated content (Altman, 2024). What impact will this have on future deep generative models and their successors? Previous work forewarned that sequences of models trained exclusively on successive iterations of their own outputs can exhibit *model collapse*: model performance degrades with each iteration and later models become progressively more unusable (Shumailov et al., 2023). Due to increasingly heavy usage of LMs to produce written content for the internet (Bommasani et al., 2022; Reuel et al., 2024; Perrault & Clark, 2024; Kapoor et al., 2024), model collapse alarmists believe current trends, if unmitigated, will irredeemably pollute the pretraining data supply and degrade language models across time.

However, the model collapse literature contains a variety of methodologies, assumptions, and models. Research supports conflicting positions about such future scenarios (Taori & Hashimoto, 2023; Hataya et al., 2023; Martínez et al., 2023; Shumailov et al., 2023; Alemohammad et al., 2024; Martínez et al., 2023; Bohacek & Farid, 2023; Guo et al., 2024; Bertrand et al., 2024; Briesch et al., 2023; Gillman et al., 2024; Wyllie et al., 2024; Dohmatob et al., 2024a;b; Gerstgrasser et al., 2024; Seddik et al., 2024; Marchi et al., 2024; Padmakumar & He, 2024; Chen et al., 2024; Ferbach et al., 2024a; Veprikov et al., 2024; Dey & Donoho, 2024). Concordance in the field is especially challenging because "model collapse" has been defined in various ways and studied under different assumptions concerning training-workflows and task-settings. Shumailov et al. (2023) defined model collapse as a "degenerative process affecting generations of learned generative models, in which the data they generate end up polluting the training set of the next generation." Dohmatob et al. (2024b) called model collapse the worsening of scaling law curves when training on synthetic as opposed to real data. In their theory sections, Shumailov et al. (2024) and Gerstgrasser et al. (2024) defined model collapse as divergent test loss after multiple iterations of training in certain task settings. Due to space constraints, we discuss Related Work more extensively in Appendix A.

In this paper, we take model collapse by its literal mean-

ing: that model performance deteriorates to the level of uselessness across iterations of an assumed data *training-workflow* under an assumed generative model *task-setting*. We bring new clarity to the discussion by studying how three different data training-workflows shape trained model performance over multiple generations. The first training-workflow, initially studied by Shumailov et al. (2023), fully *replaces* previous data at each iteration with newly generated synthetic data from the most recent generative model. In particular, under the *replace* training-workflow, real data are only used at the very first iteration. We conduct experiments in three different generative model task-settings - from basic multivariate Gaussian modeling (MGM) to kernel density estimation (KDE) to supervised fine-tuning of language models (SFT) – and observe that in each setting, the *replace* training-workflow induces collapse. We also consider the data training-workflow of *accumulating* all prior data - both real and synthetic - and find that on all three task-settings, collapse does not occur: the test loss on real data stays bounded at reasonable levels. Additionally, we introduce and study a more realistic "fixed compute" training-workflow. In this training-workflow, all real and synthetic data are retained, as in the accumulate setting. However, a dataset is subsampled from the total available pool of data to train the next model. While the pool of data grows over time, the size of the subsampled training dataset remains constant regardless of the model iteration. This training-workflow reflects the reality that available compute might not keep pace with the rate of synthetic data generation, forcing model providers to sample a fraction of future web-scale data for training. This training-workflow can be viewed as a middle ground between the assumed fixed-sample size *replace* training-workflow and the increasing sample-size *accumulate* training-workflow. We find that test errors grow larger and more quickly than in the *accumulate* scenario, but more slowly than they grow in the *replace* scenario. These results are consistent across five different generative model task-settings. Third, we investigate whether the proportion or cardinality of initial real data matters more for preventing model collapse and discover a non-trivial interaction between real and synthetic data: when real data are scarce, adding a small amount of synthetic data can reduce test loss, whereas when real data are ample, *any* synthetic data increases the test loss calculated on real data.

To summarize our contributions:

1. We show that the hypotheses of Gerstgrasser et al. (2024) are more universally applicable than originally claimed: three generative model settings widely cited as evidence for the dangers of collapse under a *replace* training-workflow (Shumailov et al., 2024) fail to collapse under the *accumulate* training-workflow.

2. We characterize test-loss behavior in a middle-ground between the replace and accumulate workflows. This middle ground reflects the impact of limited training-compute in a data ecosystem that is being flooded with synthetic data.

3. We illustrate that when real data are in limited supply, supplementing the training set by adding small amounts of synthetic data can improve test loss. On the other hand, when real data are abundant, supplementation with synthetic data only increases test loss.

## 2. Testing Two Model Collapse Claims in Three New Generative Modeling Settings

Gerstgrasser et al. (2024) recently made two claims:

1. Where model collapse has been documented, the collapse can be attributed to the *replace* workflow, where at each iteration, prior data (synthetic or real) are deleted *en masse*.

2. In the *accumulate* workflow where synthetic data accumulate alongside the original real data, model collapse is avoided.

If correct, these two claims suggest that model collapse is less likely to pose a significant threat to future deep generative models since accumulating data over time is a more realistic model of internet evolution. As a partner at Andreessen-Horowitz elegantly explained, deleting data en masse is *"not what is happening on the internet. We won't replace the Mona Lisa or Lord of the Rings with AI-generated data, but the classics will continue to be part of the training data set"* (Appenzeller, 2024). Despite such plausible heuristics, the claims of Gerstgrasser et al. (2024) were not formally tested in three generative modeling task-settings studied in influential recent work (Shumailov et al., 2024): (1) Multivariate Gaussian Modeling (MGM); (2) Kernel Density Estimation (KDE); and (3) Supervised Fine-tuning of Language Models (SFT). In this section, we ask and answer: *In these three new generative modeling settings, do Gerstgrasser et al. (2024) Claims 1 and 2 continue to hold?* In all three settings, we find that the answer is *yes*.

### 2.1. Model Collapse in Multivariate Gaussian Modeling

Following Shumailov et al. (2023); Alemohammad et al. (2024); Bertrand et al. (2024), we study what happens when one iteratively fits multivariate Gaussians and samples from them. We begin with $n$ *real* data drawn from a Gaussian with mean $\mu^{(0)}$ and covariance $\Sigma^{(0)}$: $X_1^{(0)}, ..., X_n^{(0)} \sim_{i.i.d.} \mathcal{N}(\mu^{(0)}, \Sigma^{(0)})$. For model fitting, we compute the unbiased

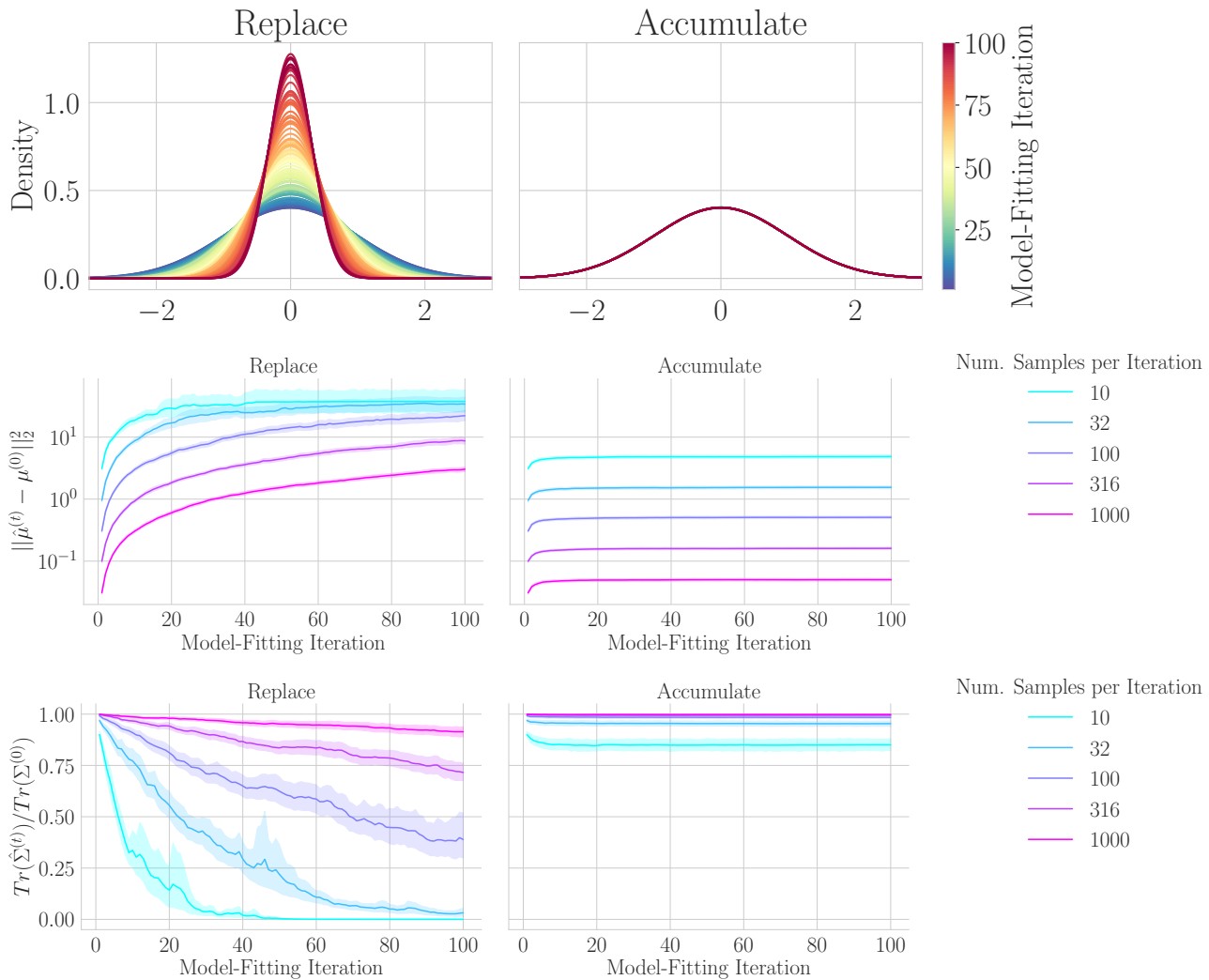

*Figure 1.* **Model Collapse in Multivariate Gaussian Modeling. Top:** Previous work (Shumailov et al., 2023; Alemohammad et al., 2024; Bertrand et al., 2024) proved model collapse occurs under the *replace* training-workflow which iteratively fits means and covariances to data, deletes earlier data, and replaces it with samples from a Gaussian with the fitted parameters (left). However, under the *accumulate* workflow where one doesn't delete data after each model-fitting iteration, model collapse does not occur (right). Note: We visualize the fit Gaussians as zero-mean for easy comparison of the fit covariances across model-fitting iterations. **Middle:** If data are replaced, then the fitted means drift away from the original data's mean, but if data instead accumulate, then the fitted means stabilize. **Bottom:** If data are replaced, then the fitted covariances collapse compared to the original data's covariance, but if past data are not discarded, the fitted covariances stabilize quickly and collapse is averted.

mean and covariance of the most recent data:

$$\hat{\mu}_{\text{Replace}}^{(t+1)} \overset{\text{def}}{=} \frac{1}{n} \sum_{j=1}^{n} X_j^{(t)}$$

$$\hat{\Sigma}_{\text{Replace}}^{(t+1)} \overset{\text{def}}{=} \frac{1}{n-1} \sum_{j=1}^{n} (X_j^{(t)} - \hat{\mu}_{\text{Replace}}^{(t+1)})(X_j^{(t)} - \hat{\mu}_{\text{Replace}}^{(t+1)})^T$$

For sampling, we draw $n$ synthetic data using the fit parameters: $X_1^{(t)}, ..., X_n^{(t)} \sim_{i.i.d.} \mathcal{N}(\hat{\mu}_{\text{Replace}}^{(t)}, \hat{\Sigma}_{\text{Replace}}^{(t)})$.

Shumailov et al. (2024) proved that as $t \to \infty$,

$$\hat{\Sigma}_{\text{Replace}}^{(t+1)} \overset{a.s.}{\to} 0$$

$$\mathbb{E}[\mathbb{W}_2^2(\mathcal{N}(\hat{\mu}_{\text{Replace}}^{(t+1)}, \hat{\Sigma}_{\text{Replace}}^{(t+1)}), \mathcal{N}(\mu^{(0)}, \Sigma^{(0)}))] \to \infty,$$

where $\mathbb{W}_2$ denotes the Wasserstein-2 distance. This result simply states that the fit covariances will collapse to 0 and that the Wasserstein-2 distance will diverge as this model-data feedback loop unfolds. However, *this result assumes that all data are replaced with new synthetic data after*

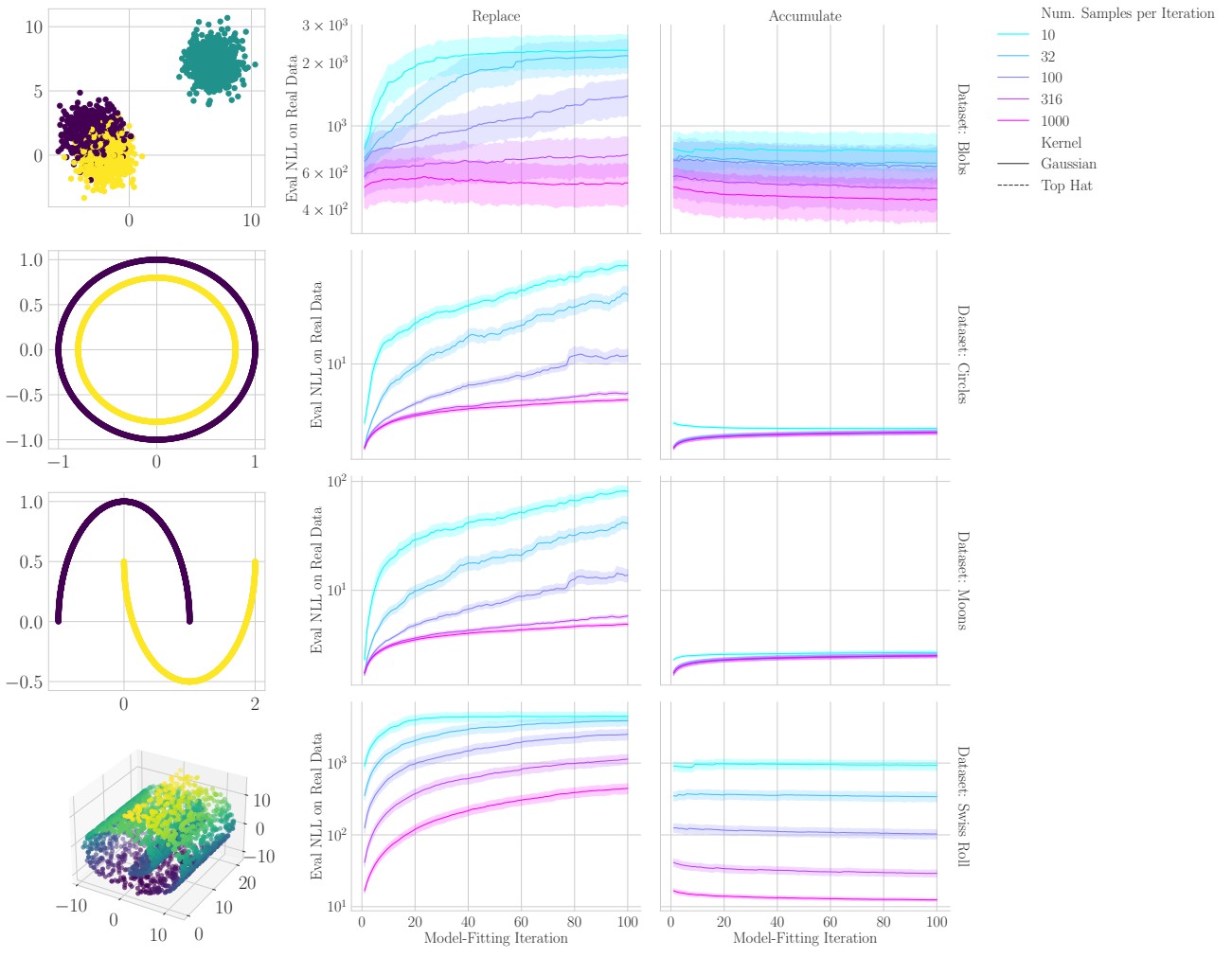

*Figure 2.* **Model Collapse in Kernel Density Estimation.** Left: We consider 4 standard datasets from `sklearn`: Blobs, Circles, Moons and Swiss Roll. Center: For all four datasets, deleting data en masse causes the negative log likelihoods (NLL) of held-out real data to increase with each model-fitting iteration. Right: For all four datasets, the *accumulate* training-workflow avoids diverging test loss on real data. Interestingly, for specific pairs of datasets and numbers of samples per iteration, training on real and accumulated synthetic data can yield *lower test-loss on held-out real data* than would training on real data alone.

*each model-fitting iteration*, which is likely unrealistic as a model of data evolution; as a rule we do not delete earlier content from the internet, replacing it en masse with new model-generated content after fitting each state-of-the-art model. To study what happens if data instead *accumulate* across model-fitting iterations, we fit to all previous real and synthetic data, a mixture in which the fraction of real data asymptotically approaches 0:

$$\hat{\mu}_{\text{Accumulate}}^{(t+1)} \stackrel{\text{def}}{=} \frac{1}{n(t+1)} \sum_{i=0}^{t} \sum_{j=1}^{n} X_j^{(i)}$$

$$\hat{\Sigma}_{\text{Accumulate}}^{(t+1)} \stackrel{\text{def}}{=} \frac{1}{n(t+1)-1} \sum_{i=0}^{t} \sum_{j=1}^{n} \bar{X}_j^{(i)} (\bar{X}_j^{(i)})^T,$$

where $\bar{X}_j^{(i)} \stackrel{\text{def}}{=} X_j^{(i)} - \hat{\mu}_{\text{Accumulate}}^{(t+1)}$ is shorthand for the centered datum. Data are then sampled using these fit Accumulate parameters rather than the fit Replace parameters.

Empirically, we find that replacing all data with new synthetic data after each model-fitting iteration causes model collapse (Fig. 1 Left), whereas accumulating data across model-fitting iterations prevents model collapse (Fig. 1 Right). More specifically, we find that if data are deleted,

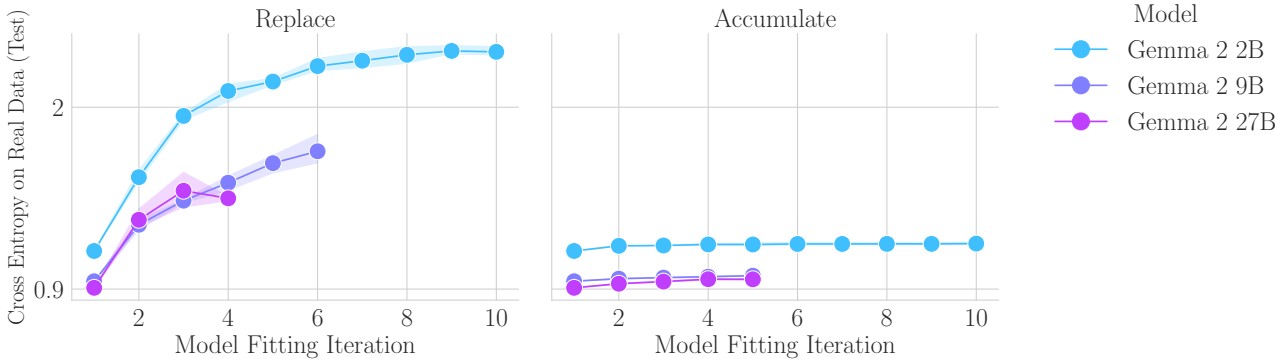

Figure 3. **Model Collapse in Supervised Fine-tuning of Language Models.** Fine-tuning Google's Gemma2 models on Nvidia's HelpSteer 2 dataset demonstrates that model collapse occurs if previous data are replaced after each model-fitting iteration (left), whereas model collapse is avoided if new synthetic data instead accumulate with previous real and synthetic data (right).

the squared error between the fit mean $\hat{\mu}^{(n)}_{\text{Replace}}$ and the initial mean $\mu^{(0)}$ diverges (Fig. 1, Middle Left), and the fit covariance $\hat{\Sigma}^{(n)}_{\text{Replace}}$ relative to the initial covariance $\Sigma^{(0)}$ collapses to 0 (Fig. 1, Bottom Left). In contrast, if data accumulate, the squared error between the fit mean and the initial mean plateaus quickly (Fig. 1, Middle Right), as does the fit covariance relative to the initial covariance (Fig. 1, Bottom Right). For univariate Gaussians, we can mathematically characterize the limit distribution:

**Theorem 1.** *For notational efficiency, for a univariate Gaussian, let $\hat{\mu}^{(t)}$ and $\hat{\sigma}^{(t)}$ denote $\hat{\mu}^{(t)}_{Accumulate}$ and $\hat{\Sigma}^{(t)}_{Accumulate}$. Then*

$$\mathbb{E}\left[\sigma_t^2\right] \quad \xrightarrow{t\to\infty} \quad \sigma_0^2 \cdot \left(\frac{\sin(\pi/\sqrt{n})}{\pi/\sqrt{n}}\right) \quad (1)$$

$$\mathbb{E}[(\mu_t - \mu_0)^2] \quad \xrightarrow{t\to\infty} \quad \sigma_0^2 \cdot \left(1 - \frac{\sin(\pi/\sqrt{n})}{\pi/\sqrt{n}}\right). \quad (2)$$

See Appendix B for the proof. This reveals two key differences when data accumulate: the covariance no longer collapses, and the mean no longer diverges. This also implies that the Wasserstein-2 distance no longer diverges. Altogether, if data accumulate, model collapse is mitigated.

### 2.2. Model Collapse in Kernel Density Estimation

Shumailov et al. (2024) introduced a second generative modeling task-setting: kernel density estimation (KDE). We begin with $n$ real data points drawn from an initial probability distribution $p^{(0)}$: $X_1^{(0)}, ..., X_n^{(0)} \sim_{i.i.d.} p^{(0)}$. We then iteratively fit KDEs to the data and sample new synthetic data from these KDEs. In the *replace* training-workflow, we fit a KDE to $n$ data from the most recently fit model, whereas in the *accumulate* training-workflow, we fit a KDE to data from all previous iterations, with the available dataset

growing linearly as $n(t + 1)$:

$$\hat{p}_{\text{Replace}}^{(t+1)}(x) \overset{\text{def}}{=} \frac{1}{nh} \sum_{j=1}^n K\left(\frac{x - X_j^{(t)}}{h}\right)$$

$$\hat{p}_{\text{Accumulate}}^{(t+1)}(x) \overset{\text{def}}{=} \frac{1}{nh(t+1)} \sum_{i=0}^t \sum_{j=1}^n K\left(\frac{x - X_j^{(i)}}{h}\right)$$

where $K$ is the kernel function and $h$ is its bandwidth parameter. We consider a standard Gaussian kernel. For sampling, at each iteration, we draw $n$ new synthetic data points from the fitted KDEs. We evaluate the performance using the negative log-likelihood (NLL) on real held-out test data; lower NLL indicates better performance. We use four standard synthetic datasets from `sklearn` (Pedregosa et al., 2011): blobs, circles, moons, and swiss roll.

Our results validate Claims 1 and 2 of Gerstgrasser et al. (2024) in the KDE task-setting. (Fig. 2): *replace* causes a rapid increase in NLL as iterations increase, indicating that the KDEs are becoming increasingly poor at modeling the real data distribution. In contrast, under *accumulate*, the NLL of real test data remains relatively stable, demonstrating that this training-workflow avoids model collapse. Surprisingly, accumulating data can yield *lower* negative log likelihoods on held-out test data that *decrease* with additional model-fitting iterations. While synthetic data are valuable elsewhere - e.g., Jain et al. (2024), Mobahi et al. (2020) - their value here is intuitive: the synthetic data "fills in" the gaps between training data points and better approximate the distribution. For the simplest 'out of the box' KDE, test losses may diverge (although very slowly) even under *accumulate*; but, as explained in the Appendices, with sufficient care in setting up the bandwidth selector of the KDE, *accumulate* test losses will not diverge. By contrast, *replace* test losses will diverge regardless of the bandwidth selector.

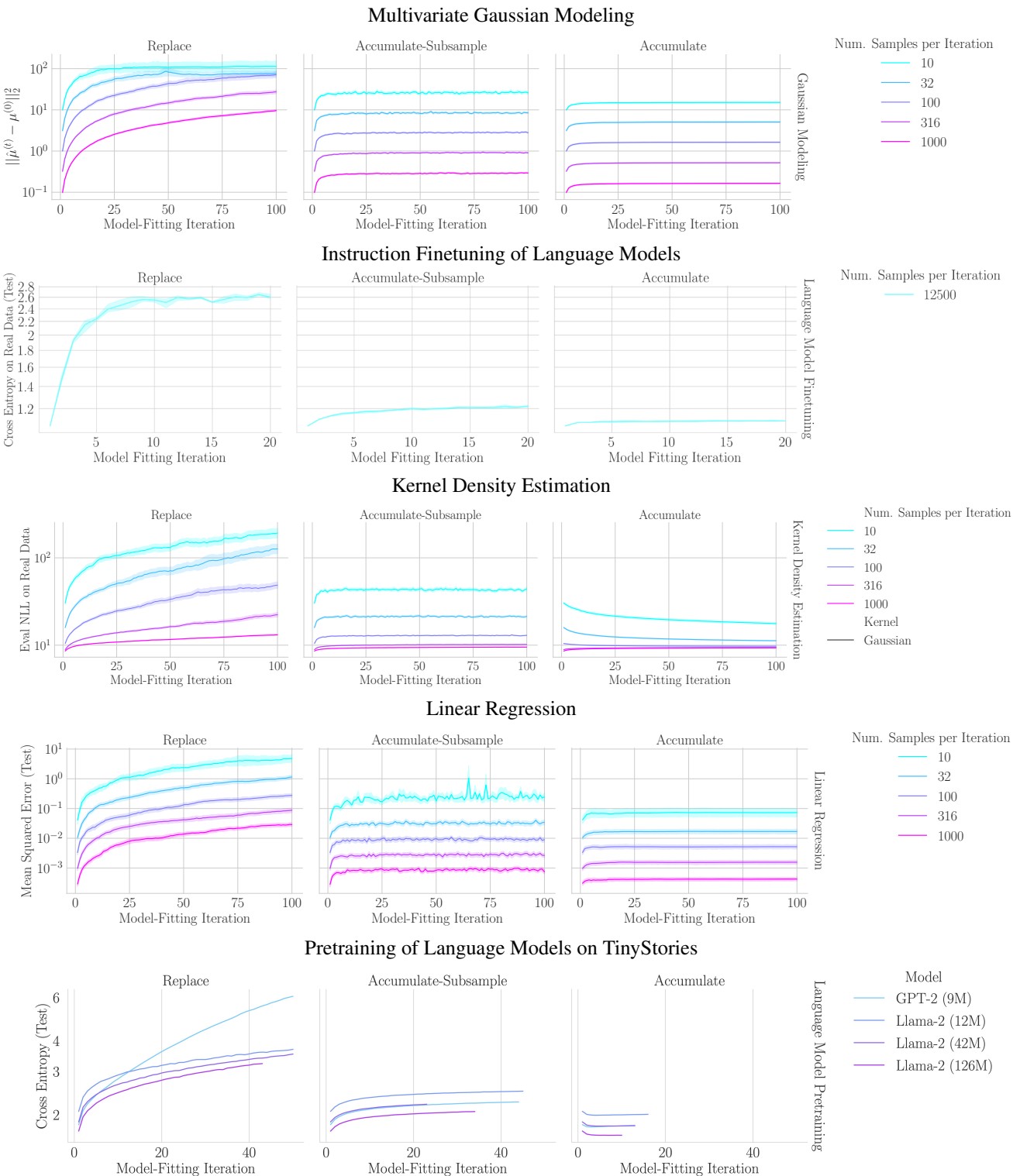

*Figure 4.* **Model Collapse Under a Fixed Compute Budget.** We compare deleting data after each model-fitting iteration (*replace*) and accumulating data after each iteration (*accumulate*) with a new fixed-compute data paradigm *accumulate-subsample*. In *accumulate-subsample*, real and synthetic data accumulate but are then subsampled so that each model is trained on a constant number of data. *Accumulate-subsample*'s test loss on real data deteriorates more quickly than *accumulate*'s loss but more slowly than *replace*'s loss, and frequently converges, albeit to a higher plateau than *accumulate*. These results hold across five task-settings: multivariate Gaussian modeling, language model instruction finetuning, kernel density estimation, linear regression and language model pretraining.

### 2.3. Model Collapse in Supervised Fine-tuning of Language Models

We now turn to the third setting for studying model collapse introduced by Shumailov et al. (2024): supervised fine-tuning of language models. Our chosen 'real data' is Nvidia's HelpSteer2 (Wang et al., 2024) instruction-following dataset; we will iteratively fine-tune a language model before sampling new text data from it. We chose Google's Gemma2 models (Team et al., 2024); they are relatively high performing and relatively small. For *replace*, we fine-tune the $t$-th language model only on data generated by the $(t-1)$ language model; for *accumulate*, we instead fine-tune the $t$-th model on the initial real data plus all synthetic data sampled from models $1, \ldots, t-1$. Thus, the amount of data for *replace* is constant $\sim 12.5\text{K}$, whereas the amount of data for *accumulate* grows linearly $\sim 12.5\text{K} \cdot t$. We again find that deleting data after each iteration leads to collapse whereas accumulating data avoids collapse (Fig. 3).

## 3. Collapse Under a Fixed Compute Budget?

Thus far, we have focused on two training-workflows: *replace* and *accumulate*. As discussed in Sec. 2, *replace* is unrealistic as a model of internet evolution: we cannot delete and regenerate the internet after pretraining each successive model. *Accumulate* might also be unrealistic because *accumulate* trains each successor model on ever-expanding data and increasing compute. For the sake of predicting likely outcomes for future generative models, we ask and answer: *Does model collapse occur when data accumulate, while each model training uses a fixed compute budget?*

We call this training-workflow *accumulate-subsample* since data accumulate but at each model-fitting iteration are subsampled to a fixed dataset size. We study the same three generative modeling task-settings as above, plus two new task-settings from other prior work (Mobahi et al., 2020; Dohmatob et al., 2024a; Gerstgrasser et al., 2024): linear regression and pretraining language models on a GPT3.5/GPT4-generated dataset of kindergarten-level text (Eldan & Li, 2023).

Across all five generative models, *accumulate-subsample*'s test loss on real data lies between the test losses of Replace and Accumulate (Fig. 4). Specifically, *accumulate-subsample* exhibits higher test loss than *accumulate* but lower test loss than *replace*, showing that the fixed compute budget imposes some performance penalty. In a qualitative difference, test losses on real data typically plateau for both *accumulate-subsample* and *accumulate*, while test losses for *replace* typically diverge in an apparently unbounded manner. Thus, modifying *accumulate* to the more compute-realistic workflow *accumulate-subsample*, the threat of model collapse is still avoided.

## 4. Cardinality of Real Data vs Proportion of Real Data in Mitigating Model Collapse

We conclude by turning to a key open question: *Which matters more for avoiding model collapse: the cardinality of real data or the proportion of real data? Relatedly, how does the value of synthetic data for reducing test loss on real data depend on the amount of real data?*

These questions are highly pertinent to researchers sampling from web-scale data in order to pretrain or finetune language models. To explore these questions, we fine-tune Google's Gemma 2 2B model using the HelpSteer2 dataset. We generate 100k completions from the fine-tuned model and filter out those exceeding 512 tokens, leaving 55,000 usable samples. We then construct datasets with varying mixes of real and synthetic data and fine-tune Gemma 2B on each. The final test loss for each configuration is recorded (Fig. 5).

This experiment provides several insights. First, both the number and proportion of real data have an impact on the test loss following SFT. To assess this, we first transformed the number of real datapoints $n$ as $\frac{1}{n^{1/2}}$, in keeping with intuitions from classical statistics on how the log likelihood scales with the number of data points. Then, based on observation of the data, we computed

$$\log\left(\frac{\text{real data}}{\text{real data} + \text{synthetic data}}\right)$$

to best capture the relationship between the fraction of real data and the log likelihood. We measured $R^2$ values of $0.59$ for the transformed number of real data and $0.34$ for the proportion of real data. We next computed $F$-statistics for the one-term versus two-term models involving each of these covariates, which gave us $p$-values of $6.9 \times 10^{-25}$ and $4.6 \times 10^{-25}$, respectively. These statistics suggest that both the proportion and the cardinality of real data have a statistically significant effect on the test loss, and explain a sizable fraction of the variance in the test loss.

Second, we find a difference in the effect of synthetic data on test loss comparing high- versus low- real-data regimes. In our experiments, when the number of real data is 1024 or lower, we find that there is a *small but non-zero amount of synthetic data that improves the test loss when it is included*. This optimal number of synthetic datapoints appears to be near 1024. This suggests that practitioners fine-tuning with insufficient amounts of real data should consider supplementing with synthetic data to improve model quality. On the other hand, when real data are plentiful (more than 1024 datapoints), we find that more synthetic data almost always degrades final model quality when the number of real data is held constant. When there are more than 1024 real datapoints, datasets containing only real data prove to be more valuable than datasets that contain ten times more real data mixed with synthetic data.

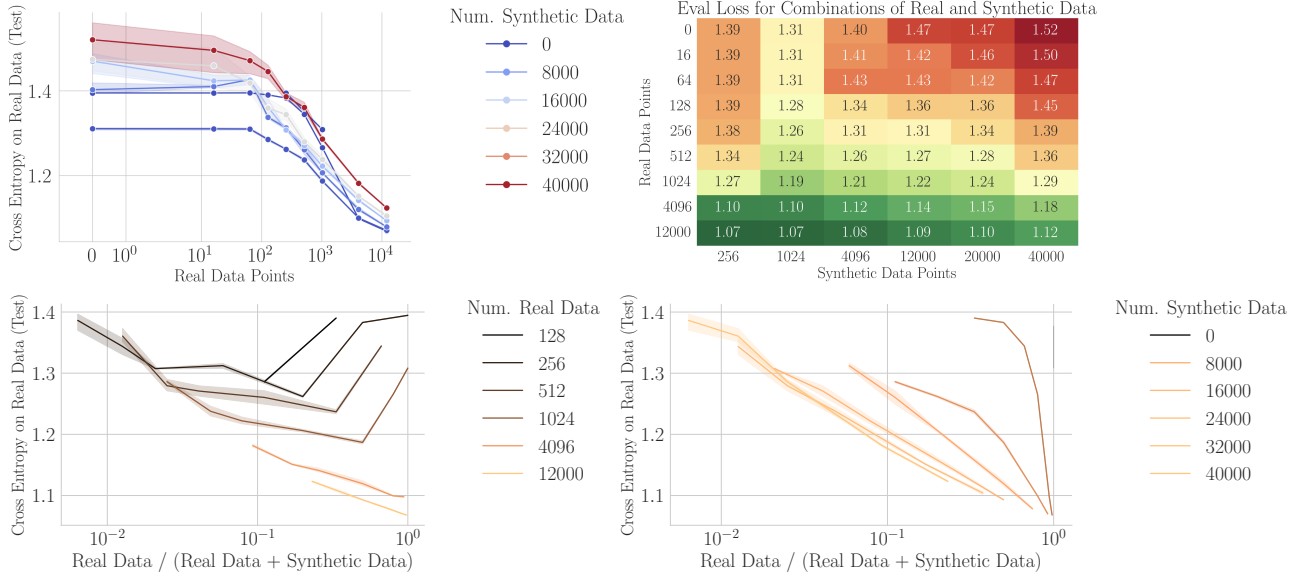

*Figure 5.* **The Value of Synthetic Data in Supervised Fine-tuning of Language Models.** Fine-tuning Google's Gemma 2 2B on Nvidia's HelpSteer 2 dataset on different combinations of real and synthetic data. We observe that when the number of real data is small, supplementing with synthetic data can improve test loss. With sufficient real data, adding synthetic data degrades performance.

These results raise interesting questions about the role of synthetic data in SFT that merit exploration. We can sometimes achieve better results by removing *all* synthetic data from the training set than by doubling the amount of real data. When constructing datasets subject to cost constraints, these results suggest that removing synthetic or low-quality data can sometimes bring more value than collecting greater volumes of high-quality data. These experimental results bear comparison with prior theoretical work of Dohmatob et al. (2024b), cf. Corollary 3.3 et seq.

## 5. Discussion

We studied the risks of model collapse and identified two pathways to containment. We demonstrated in three new generative modeling settings that accumulating data over time avoids model collapse, whereas replacing data over time induces model collapse. In five generative modeling settings, we demonstrated that using a modified accumulate workflow - where each model trains on a fixed-size sample drawn from all available real and synthetic data - model performance deteriorates, but still tends to plateau. The consistency of results across model types and datasets suggests that *this distinction is a general phenomenon, and is not specific to any particular model, dataset, or learning algorithm.* Lastly, we explored the value of synthetic data for reducing the test loss on real data and found two different regimes: when real data are plentiful, adding synthetic data can be harmful, but when real data are scarce, there exists

an optimal amount of synthetic data that should be added.

To us, the most realistic viewpoint on internet evolution assumes synthetic data accumulating from a host of models alongside continuing influx of real-world data. Combining it with our experimental and analytical work, model collapse seems unlikely. Our experiments take a pessimistic viewpoint, in the sense that our experiments pay no attention to the quality of data, whereas in practice, engineers heavily filter data based on various indicators of data quality, e.g., (Brown et al., 2020; Lee et al., 2023; Wettig et al., 2024; Penedo et al., 2024; Li et al., 2024b; Sachdeva et al., 2024); for a recent review, see Albalak et al. (2024).

A promising future direction might combine synthetic data generation with filtering techniques to enable performant and efficient pretraining at scale. As we saw in kernel density estimation (Fig. 2) and in language model pretraining on TinyStories (Fig. 4), training on real and synthetic data accumulated over several iterations can yield lower loss on real test data than training on real data alone. Identifying under what conditions synthetic data can lower test loss during pretraining would be invaluable to industry practitioners.

Our results in Section 4 suggest that removing low-quality synthetic data from model training sets *can improve test loss more than gathering additional high-quality data.* Developing efficient identification and removal techniques for detrimental data could streamline the model fine-tuning process and produce better alignment.

## Acknowledgements

Redacted for double blind reviewer.

## Impact Statement

This paper advances the understanding of model collapse, a critical issue in the era of synthetic data proliferation. By systematically analyzing different data evolution paradigms, our findings provide actionable insights into mitigating the risks associated with training generative models on recursively generated data.

The societal implications of our work are substantial. If left unaddressed, model collapse could lead to the degradation of AI-generated content, reducing the reliability of machine learning models deployed across industries, including healthcare, finance, and education. Our results highlight that while synthetic data can be beneficial under controlled accumulation, indiscriminate use may lead to severe model degradation. This has ethical ramifications for AI deployment, as biased or deteriorating models could reinforce misinformation, amplify biases, or reduce transparency in decision-making systems.

Additionally, our study underscores the importance of responsible data curation and compute-aware training strategies. As AI systems become increasingly self-referential, policymakers and practitioners must develop strategies to balance synthetic and real data, ensuring sustainable model performance. Our work informs these discussions by demonstrating that synthetic data, if managed appropriately, can enhance rather than harm AI models.

While our findings primarily contribute to advancing machine learning theory, we encourage continued research into the ethical governance of AI-generated datasets, particularly as reliance on synthetic data expands. Future work should explore mechanisms for filtering low-quality synthetic data and designing robust training paradigms to prevent long-term degradation of AI capabilities.

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

# A. Related Work

The limitations of using AI-generated images to train other image models have been well-documented since 2022 (Hataya et al., 2023). Shumailov et al. (2023) most prominently sounded the alarm about synthetic data for training language models by showing that a model trained repeatedly on its own outputs exhibits severely denigrated quality. Shumailov's theory and empirical work were quickly extended to many new settings (Alemohammad et al., 2024; Bertrand et al., 2024; Dohmatob et al., 2024b;a; Marchi et al., 2024).

Within the model collapse literature, a variety of data dynamics have been studied, which vary in how "real" data are discarded or retained, how "synthetic" data are generated, and how each is (or is not) incorporated into future training sets (Martínez et al., 2023; Mobahi et al., 2020; Dohmatob et al., 2024a). A common feature of many of these is that at least some real data are discarded, often because total dataset size is kept constant across model-fitting iterations. However, (Gerstgrasser et al., 2024) note that this may not represent real-world dynamics, and that model collapse is avoided when data accumulate. What is not clear, however, is whether this claim holds universally, including in the specific settings studied in other prior work. We help close this gap by extending Gerstgrasser's empirical and theoretical analyses to several of these settings.

Where model collapse can be seen as studying a worst-case scenario, it has also been observed that *some* kinds of synthetic data have a positive effect. (Dohmatob et al., 2024b) and (Jain et al., 2024) find that certain amounts of synthetic data can improve model performance, and (Ferbach et al., 2024b) suggest that with curation, self-consuming loops can improve alignment with human preferences. A growing literature on how to filter and harness synthetic data has achieved impressive results on a variety of benchmarks (Zelikman et al., 2024; Li et al., 2024a; Yang et al., 2024), raising interesting questions about the limits of when unfiltered synthetic data can help. In this vein, we answer a question posed by (Gerstgrasser et al., 2024): does the proportion or the raw amount of real data in a mixed training set have a greater impact on test loss? In the process, we find that proportionally small amounts of synthetic data can improve test loss when real data is scarce.

## B. Iterative Gaussian Model Fitting: Mathematical Results and Proofs

### B.1. Setup

**Lemma 2.** *Using the notation of Theorem 1, we can express $\mu_t = \sum_{r=1}^{t} \sigma_{r-1} \frac{\overline{z_r}}{r} + \mu_0$.*

*Proof.* Note that $X_{i,t} = \mu_{t-1} + \sigma_{t-1} z_{i,t}$, where $z_{i,t} \sim \mathcal{N}(0,1)$. Therefore,

$$
\begin{aligned}
\mu_t &= \frac{1}{nt} \sum_{r=1}^{t} \sum_{i=1}^{n} X_{i,r} \\
&= \frac{t-1}{t} \mu_{t-1} + \frac{\mu_{t-1}}{t} + \sigma_{t-1} \frac{\overline{z_t}}{t} \\
&= \mu_{t-1} + \sigma_{t-1} \frac{\overline{z_t}}{t}.
\end{aligned}
$$

Therefore, $\mu_t = \sum_{r=1}^{t} \sigma_{r-1} \cdot \frac{\overline{z_r}}{r} + \mu_0$. $\qquad\square$

**Lemma 3.** *Under the setup described in Theorem 1, $\mathbb{E}[\frac{\sigma_t^2}{\sigma_0^2}] = \prod_{k=1}^{t} \left(1 - \frac{1}{nk^2}\right) \xrightarrow{t \to \infty} \frac{\sin(\pi/\sqrt{n})}{\pi/\sqrt{n}}$.*

*Proof.* Using the recursive expression for $\mu_t$ in Lemma 2, we can rewrite

$$
\begin{aligned}
\sigma_t^2 &= \frac{1}{nt} \sum_{r=1}^{t} \sum_{i=1}^{n} (X_{i,r} - \mu_t)^2 \\
&= \frac{1}{nt} \sum_{r=1}^{t} \sum_{i=1}^{n} \left(X_{i,r} - \overline{X_r} + \overline{X_r} - \mu_t\right)^2 \\
&= \frac{1}{nt} \sum_{r=1}^{t} \left(\sum_{i=1}^{n} \left(X_{i,r} - \overline{X_r}\right)^2 + n(\overline{X_r} - \mu_t)^2\right) \\
&= \frac{1}{t} \sum_{r=1}^{t} \left(\sigma_{r-1}^2 S_r^2 + (\mu_{r-1} + \sigma_{r-1} \overline{z_r} - \mu_t)^2\right).
\end{aligned}
$$

In the last line, we define $S_r^2 = \sum_{i=1}^{n} (z_{i,r} - \overline{z_r})^2$. The term

$$
(\mu_{r-1} + \sigma_{r-1} \overline{z_r} - \mu_t)^2 = \left(\sigma_{r-1} \overline{z_r} - \sum_{k=r}^{t} \sigma_{k-1} \cdot \frac{\overline{z_k}}{k}\right)^2,
$$

so

$$
\begin{aligned}
\sigma_t^2 &= \frac{1}{t} \sum_{r=1}^{t} \left(\sigma_{r-1}^2 S_r^2 + \left(\sigma_{r-1} \overline{z_r} - \sum_{k=r}^{t} \sigma_{k-1} \frac{\overline{z_k}}{k}\right)^2\right) \\
\Rightarrow t\sigma_t^2 &= \sum_{r=1}^{t} \left(\sigma_{r-1}^2 S_r^2 + \left(\sigma_{r-1} \overline{z_r} \left(1 - \frac{1}{r}\right) - \sum_{k=r+1}^{t} \sigma_{k-1} \frac{\overline{z_k}}{k}\right)^2\right).
\end{aligned}
$$

We now compute the conditional expectations of the terms in this sum. Where $\mathcal{F}_i$ denotes the $i$th filtration,

$$
\mathbb{E}[\sigma_{r-1}^2 S_r^2 | \mathcal{F}_{t-1}] = \begin{cases} \sigma_{r-1}^2 S_r^2 & r < t \\ \sigma_{t-1}^2 \cdot \left(\frac{n-1}{n}\right) & r = t. \end{cases}
$$

For $r = t$, we find that

$$\mathbb{E}\left[\left(\sigma_{r-1}\overline{z_r} \cdot \left(1 - \frac{1}{r}\right) - \sum_{k=r+1}^{t} \sigma_{k-1} \cdot \frac{\overline{z_k}}{k}\right)^2 \Big| \mathcal{F}_{t-1}\right] = \sigma_{t-1}^2 \left(1 - \frac{1}{t}\right) \cdot \frac{1}{n}.$$

On the other hand, when $r < t$,

$$\mathbb{E}\left[\left(\sigma_{r-1}\overline{z_r} \cdot \left(1 - \frac{1}{r}\right) - \sum_{k=r+1}^{t-1} \sigma_{k-1} \cdot \frac{\overline{z_k}}{k} - \sigma_{t-1} \cdot \frac{\overline{z_t}}{t}\right)^2 \Big| \mathcal{F}_{t-1}\right]$$

$$= \sigma_{t-1}^2 \cdot \frac{1}{t^2} \cdot \frac{1}{n} + \left(\sigma_{r-1}\overline{z_r} \cdot \left(1 - \frac{1}{r}\right) - \sum_{k=r+1}^{t-1} \sigma_{k-1} \cdot \frac{\overline{z_k}}{k}\right)^2.$$

Therefore,

$$\mathbb{E}[t\sigma_t^2|\mathcal{F}_{t-1}] = (t-1)\sigma_{t-1}^2 + \sigma_{t-1}^2 \cdot \left(1 - \frac{1}{n}\right) + \sigma_{t-1}^2 \cdot \left(\frac{t-1}{t}\right) \cdot \left(\frac{1}{n}\right) + \sigma_{t-1}^2 \cdot \left(1 - \frac{1}{t}\right)^2 \cdot \left(\frac{1}{n}\right)$$

$$= \sigma_{t-1}^2 \left(t - 1 + 1 - \frac{1}{n} + \frac{1}{tn} - \frac{1}{t^2 n} + \frac{1}{n} - \frac{2}{tn} + \frac{1}{t^2 n}\right)$$

$$= \sigma_{t-1}^2 \left(t - \frac{1}{tn}\right).$$

It follows that

$$\mathbb{E}[\sigma_t^2|\mathcal{F}_{t-1}] = \sigma_{t-1}^2 \left(1 - \frac{1}{t^2 n}\right) < \sigma_{t-1}^2$$

for all $t$. Thus, $\{\sigma_t^2\}_t$ is a supermartingale, and

$$\sigma_t^2 \xrightarrow{a.s.} \sigma_\infty^2$$

because $\sigma_t^2$ is bounded below by $0$. Therefore, we still have convergence. Next, letting $m_t = \mathbb{E}[\sigma_t^2]$, we have

$$m_t = m_{t-1}\left(1 - \frac{1}{t^2 n}\right) = \cdots = \sigma_0^2 \prod_{k=1}^{t}\left(1 - \frac{1}{k^2 n}\right),$$

so

$$\mathbb{E}[\sigma_t^2] = \sigma_0^2 \prod_{k=1}^{\infty}\left(1 - \frac{1}{k^2 n}\right). \tag{3}$$

By a theorem of Euler, this is equal to

$$\sigma_0^2 \frac{\sin(\pi/\sqrt{n})}{\pi/\sqrt{n}}. \tag{4}$$

$\square$

Observe that by performing a variable replacement and using L'Hospital's rule, it is clear that $\lim_{n\to\infty} \mathbb{E}[\sigma_t^2] = \sigma_0^2$.

Finally, we are able to compute $\mathbb{E}[(\mu_t - \mu_0)^2]$.

**Corollary 4.** *The expected error in the mean*

$$\mathbb{E}[(\mu_t - \mu_0)^2] = \sigma_0^2 \left(1 - \prod_{k=1}^{t}\left(1 - \frac{1}{k^2 n}\right)\right). \tag{5}$$

*Proof.* Using the recursion from Lemma 2 and the expression for the variance in Lemma 3, we can rewrite

$$
\begin{aligned}
\mathbb{E}[(\mu_t - \mu_0)^2] &= \sum_{k=1}^{t} \frac{\mathbb{E}[\sigma_{k-1}^2]}{nk^2} \\
&= \sigma_0^2 \sum_{k=1}^{t} \frac{1}{k^2 n} \prod_{\ell=1}^{k-1} \left(1 - \frac{1}{\ell^2 n}\right) \\
&= \sigma_0^2 \sum_{k=1}^{t} \left(\prod_{\ell=1}^{k-1} \left((1 - \frac{1}{\ell^2 n})\right) - \prod_{\ell=1}^{k} \left(1 - \frac{1}{\ell^2 n}\right)\right) \\
&= \sigma_0^2 \left(1 - \prod_{k=1}^{t} \left(1 - \frac{1}{k^2 n}\right)\right).
\end{aligned}
$$

$\square$

Therefore,

$$
\lim_{t \to \infty} \mathbb{E}[(\mu_t - \mu_0)^2] = \sigma_0^2 \left(1 - \frac{\sin(\pi/\sqrt{n})}{\pi/\sqrt{n}}\right).
$$

## C. Iterative KDE Fitting: Mathematical Results and Proofs

In this section, we prove that the NLL diverges when iteratively fitting KDE's regardless of whether one accumulates or replaces data from previous iterations.

**Theorem 5.** *In the replace setting described in Section 2.2, as long as one holds the bandwidth constant, the NLL asymptotically diverges.*

*Proof.* Define $f_0$ as the density function for the data distribution from which the original data $x_1, ..., x_n$ are sampled. Define $K_h$ to be the Gaussian kernel function with fixed bandwidth $h$. One can rewrite the fitted distribution at iteration $t$ as

$$
D_t = K_h * D_{t-1}
$$

where $*$ denotes the standard convolution of densities.

By a simple recursion, it is clear that $D_t = K^{*t} * D_0$. When two Gaussian kernels with bandwidths $a$ and $b$ are convolved, a basic calculation shows that the resulting effective bandwidth is $\sqrt{a^2 + b^2}$. Consequently, by an inductive argument, the effective bandwidth of $K^{*t}$ is $h\sqrt{t}$. Therefore,

$$
\lim_{t \to \infty} K^{*t} * D_0 = \lim_{t \to \infty} K_{h\sqrt{t}} * D_0 = 0
$$

because as the bandwidth goes to $\infty$, the likelihood of any point goes to $0$. Hence, regardless of the choice of test data, the negative log likelihood diverges to $-\infty$. $\square$

The same conclusion holds when one accumulates rather than subsampling data:

**Theorem 6.** *For any non-trivial kernel (i.e. a kernel whose Fourier transform is not 1), 2.2, the NLL diverges.*

*Proof.* We adopt the same notation as in Theorem 5, except this time $K$ denotes a general kernel $K$ that doesn't necessarily need to be Gaussian. In this instance, it is more convenient to work in frequency space, where convolution in probability space corresponds to multiplication.

Define $\varphi_0$ as the Fourier transform (FT) of $f_0$, also called the characteristic function. Let $\kappa$ denote the FT of $K$. Then

$$
\varphi_t = \kappa \cdot \varphi_{t-1}
$$

where $\cdot$ denotes standard complex multiplication. Define $\delta_t = \frac{\phi_t}{\phi_0}$ so that $\varphi_t = \delta_t \cdot \varphi_0$. Define $d_t = \varphi_t / \varphi_0$, and let $a_t = \frac{1}{t} \sum_{i=0}^t d_i$. Using this notation,

$$d_t = \kappa \cdot a_{t-1} \tag{6}$$
$$a_t = ((t-1)a_{t-1} + d_t)/t. \tag{7}$$

We see that $a_t = L_{t,K}(a_{t-1})$ is an affine map with slope $((t-1) + \kappa)/t$ and intercept 0. Suppose that the characteristic function of the density converges to $\varphi_\infty$. Then the map $a_t$ has a fixed point. As long as $\kappa \neq 1$, this fixed point must satisfy the equation

$$\varphi = ((t-1) + \kappa)\varphi$$
$$\Rightarrow 0 = ((t-1) + \kappa)/g - 1) \varphi$$
$$\Rightarrow 0 = (-1 + \kappa) \varphi \Rightarrow \varphi = 0.$$

Note that if $\varphi_\infty = 0$, its inverse FT is a function that has 0 probability density everywhere in probability space. Equivalently, the variance of $f_t$ diverges to $\infty$.

$\square$

Although the NLL eventually diverges in the accumulate case, it is clear from the expression for $a_t$ that this divergence occurs very slowly.

For a Gaussian kernel, both the replace and accumulate case offer an interesting shared insight. Throughout the iterative fitting process, regardless of whether we accumulate or replace, the bandwidth monotonically grows. Therefore, when one starts this process with a very small bandwidth smaller than the optimal bandwidth for the density being fit, one could initially observe a decrease in the negative log likelihood as the bandwidth approaches its optimum.

Finally, model collapse, while inevitable with a fixed bandwidth, can be avoided in all cases by shrinking the bandwidth at a sufficiently fast rate. Since practitioners typically optimize their bandwidth according to the amount of the data that they have, the bandwidth should have the form $c(tn)^{1/5}$ where $c$ is a constant. In this setting, model collapse is avoided entirely.

**Theorem 7.** *Under the accumulate training workflow consider density estimation as in Section 2.2 with a Gaussian kernel. Let the bandwidth at the $t$th model-fitting iteration be $c \cdot (tn)^{-1/5}$ for a constant $c$. Then the asymptotic variance of the limiting KDE is finite.*

*Proof.* Let $K_{c(tn)^{-1/5}}$ denote the kernel at the $t$th model-fitting iteration. Let $f_0$ denote the original distribution, and define $f_t$ to be the distribution of the KDE at the $t$th iteration.

We can write

$$f_t = \frac{1}{t} \sum_{i=1}^t f_{i-1} * K_{c(in)^{-1/5}}$$
$$= \left(1 - \frac{1}{t}\right) \cdot \left(\frac{1}{t-1} \sum_{i=1}^{t-1} f_{i-1} * K_{c(in)^{-1/5}}\right) + \frac{1}{t} f_{t-1} * K_{c(tn)^{-1/5}}$$
$$= \left(1 - \frac{1}{t}\right) f_{t-1} + \frac{1}{t} f_{t-1} * K_{c(tn)^{-1/5}}$$
$$= \left(\left(1 - \frac{1}{t}\right) K_0 + \frac{1}{t} K_{c(tn)^{-1/5}}\right)$$

where $K_0$ is the identity kernel, or equivalently the Gaussian kernel with 0 bandwidth.

Therefore, we find that

$$f_t = f_0 * \circledast_{i=1}^t \left(\left(1 - \frac{1}{i}\right) K_0 + \frac{1}{i} K_{c(in)^{-1/5}}\right).$$

Define $W_i$ to be a random variable that is $K_{c(in)^{-1/5}}$ with probability $\frac{1}{i}$ and $K_0$ with probability $1 - \frac{1}{i}$. We can rewrite $X_t$, a random variable drawn at the $t$th fitting iteration as

$$X_t = X_0 + \sum_{i=1}^{t} W_i.$$

All of $X_0, W_1, ..., W_t$ are independent. The variance is given by

$$\mathrm{Var}(X_t) = \mathrm{Var}(X_0) + \sum_{i=1}^{t} \mathrm{Var}(W_i)$$

$$= \mathrm{Var}(X_0) + \sum_{i=1}^{t} \frac{1}{i} \times \frac{c}{(in)^{2/5}}$$

$$= \mathrm{Var}(X_0) + \frac{c}{n^{2/5}} \sum_{i=1}^{t} \frac{1}{i^{7/5}}.$$

As $t \to \infty$,

$$\mathrm{Var}(X_t) \to \mathrm{Var}(X_0) + \frac{c}{n^{2/5}} \sum_{i=1}^{\infty} \frac{1}{i^4} < \infty.$$

Therefore, when the kernel size is appropriately adjusted, the variance of the KDE under accumulate converges. $\qquad \square$

# D. Experimental Results: Sweep Configurations

## D.1. Model Collapse in Multivariate Gaussian Modeling

To study model collapse in multivariate Gaussian modeling, we ran the following YAML sweep:

```
program: src/fit_gaussians/fit_gaussians.py
project: rerevisiting-model-collapse-fit-gaussians
method: grid
parameters:
  data_dim:
    values: [ 1, 3, 10, 31, 100 ]
  num_samples_per_iteration:
    values: [10, 32, 100, 316, 1000]
  num_iterations:
    values: [ 100 ]
  seed:
    values: [ 0, 1, 2, 3, 4, 5, 6, 7, 8, 9, 10, 11, 12, 13, 14,
    15, 16, 17, 18, 19, 20, 21, 22, 23, 24, 25, 26, 27, 28, 29,
    30, 31, 32, 33, 34, 35, 36, 37, 38, 39, 40, 41, 42, 43, 44,
    45, 46, 47, 48, 49, 50, 51, 52, 53, 54, 55, 56, 57, 58, 59,
    60, 61, 62, 63, 64, 65, 66, 67, 68, 69, 70, 71, 72, 73, 74,
    75, 76, 77, 78, 79, 80, 81, 82, 83, 84, 85, 86, 87, 88, 89,
    90, 91, 92, 93, 94, 95, 96, 97, 98, 99 ]
  setting:
    values: [
      "Accumulate",
      "Accumulate-Subsample",
      "Replace",
    ]
  sigma_squared:
    values: [
      1.0,
    ]
```

Seeds were swept from 0 to 99, inclusive.

## D.2. Model Collapse in Kernel Density Estimation

To study model collapse in kernel density estimation, we ran the following YAML sweep:

```
program: src/fit_kdes/fit_kdes.py
project: rerevisiting-model-collapse-fit-kdes
method: grid
parameters:
  data_config:
    parameters:
      dataset_name:
        values: ["blobs"]
      dataset_kwargs:
        parameters:
          n_features:
            values: [2]
  kernel:
    values: ["gaussian"]
  kernel_bandwidth:
    values: [0.1, 0.5, 1.0]
  num_samples_per_iteration:
    values: [10, 32, 100, 316, 1000]
  num_iterations:
    values: [ 100 ]
```

```
seed :
  values : [ 0, 1, 2, 3, 4, 5, 6, 7, 8, 9, 10, 11, 12, 13, 14,
    15, 16, 17, 18, 19, 20, 21, 22, 23, 24, 25, 26, 27, 28, 29,
    30, 31, 32, 33, 34, 35, 36, 37, 38, 39, 40, 41, 42, 43, 44,
    45, 46, 47, 48, 49, 50, 51, 52, 53, 54, 55, 56, 57, 58, 59,
    60, 61, 62, 63, 64, 65, 66, 67, 68, 69, 70, 71, 72, 73, 74,
    75, 76, 77, 78, 79, 80, 81, 82, 83, 84, 85, 86, 87, 88, 89,
    90, 91, 92, 93, 94, 95, 96, 97, 98, 99 ]
setting :
  values : [
    "Accumulate",
    "Accumulate-Subsample",
    "Replace",
  ]

program : src/fit_kdes/fit_kdes.py
project : rerevisiting-model-collapse-fit-kdes
method : grid
parameters :
  data_config :
    parameters :
      dataset_name :
        values : ["circles"]
      dataset_kwargs :
        parameters :
          noise :
            values : [0.05]
  kernel :
    values : ["gaussian"]
  kernel_bandwidth :
    values : [0.1, 0.5, 1.0]
  num_samples_per_iteration :
    values : [10, 32, 100, 316, 1000]
  num_iterations :
    values : [ 100 ]
  seed :
    values : [ 0, 1, 2, 3, 4, 5, 6, 7, 8, 9, 10, 11, 12, 13, 14,
      15, 16, 17, 18, 19, 20, 21, 22, 23, 24, 25, 26, 27, 28, 29,
      30, 31, 32, 33, 34, 35, 36, 37, 38, 39, 40, 41, 42, 43, 44,
      45, 46, 47, 48, 49, 50, 51, 52, 53, 54, 55, 56, 57, 58, 59,
      60, 61, 62, 63, 64, 65, 66, 67, 68, 69, 70, 71, 72, 73, 74,
      75, 76, 77, 78, 79, 80, 81, 82, 83, 84, 85, 86, 87, 88, 89,
      90, 91, 92, 93, 94, 95, 96, 97, 98, 99 ]
  setting :
    values : [
      "Accumulate",
      "Accumulate-Subsample",
      "Replace",
    ]

program : src/fit_kdes/fit_kdes.py
project : rerevisiting-model-collapse-fit-kdes
method : grid
parameters :
```

```
data_config:
  parameters:
    dataset_name:
      values: ["moons"]
    dataset_kwargs:
      parameters:
        noise:
          values: [0.05]
kernel:
  values: ["gaussian"]
kernel_bandwidth:
  values: [0.1, 0.5, 1.0]
num_samples_per_iteration:
  values: [10, 32, 100, 316, 1000]
num_iterations:
  values: [ 100 ]
seed:
  values: [ 0, 1, 2, 3, 4, 5, 6, 7, 8, 9, 10, 11, 12, 13, 14,
    15, 16, 17, 18, 19, 20, 21, 22, 23, 24, 25, 26, 27, 28, 29,
    30, 31, 32, 33, 34, 35, 36, 37, 38, 39, 40, 41, 42, 43, 44,
    45, 46, 47, 48, 49, 50, 51, 52, 53, 54, 55, 56, 57, 58, 59,
    60, 61, 62, 63, 64, 65, 66, 67, 68, 69, 70, 71, 72, 73, 74,
    75, 76, 77, 78, 79, 80, 81, 82, 83, 84, 85, 86, 87, 88, 89,
    90, 91, 92, 93, 94, 95, 96, 97, 98, 99 ]
setting:
  values: [
    "Accumulate",
    "Accumulate-Subsample",
    "Replace",
  ]

program: src/fit_kdes/fit_kdes.py
project: rerevisiting-model-collapse-fit-kdes
method: grid
parameters:
  data_config:
    parameters:
      dataset_name:
        values: ["swiss_roll"]
      dataset_kwargs:
        parameters:
          noise:
            values: [0.05]
  kernel:
    values: ["gaussian"]
  kernel_bandwidth:
    values: [0.1, 0.5, 1.0]
  num_samples_per_iteration:
    values: [10, 32, 100, 316, 1000]
  num_iterations:
    values: [ 100 ]
  seed:
    values: [ 0, 1, 2, 3, 4, 5, 6, 7, 8, 9, 10, 11, 12, 13, 14,
      15, 16, 17, 18, 19, 20, 21, 22, 23, 24, 25, 26, 27, 28, 29,
```

```
   30, 31, 32, 33, 34, 35, 36, 37, 38, 39, 40, 41, 42, 43, 44,
   45, 46, 47, 48, 49, 50, 51, 52, 53, 54, 55, 56, 57, 58, 59,
   60, 61, 62, 63, 64, 65, 66, 67, 68, 69, 70, 71, 72, 73, 74,
   75, 76, 77, 78, 79, 80, 81, 82, 83, 84, 85, 86, 87, 88, 89,
   90, 91, 92, 93, 94, 95, 96, 97, 98, 99 ]
 setting:
   values: [
     "Accumulate",
     "Accumulate-Subsample",
     "Replace",
   ]
```

Seeds were swept from 0 to 99, inclusive.

### D.3. Model Collapse in Linear Regression

To study model collapse in linear regression, we ran the following YAML sweep:

```
program: src/fit_linear_regressions/fit_linear_regressions.py
project: rerevisiting-model-collapse-fit-lin-regr
method: grid
parameters:
 data_dim:
   values: [ 100, 10, 31, 3, 1 ]
 num_samples_per_iteration:
   values: [10, 32, 100, 316, 1000]
 num_iterations:
   values: [ 100 ]
 seed:
   values: [ 0, 1, 2, 3, 4, 5, 6, 7, 8, 9, 10, 11, 12, 13, 14,
   15, 16, 17, 18, 19, 20, 21, 22, 23, 24, 25, 26, 27, 28, 29,
   30, 31, 32, 33, 34, 35, 36, 37, 38, 39, 40, 41, 42, 43, 44,
   45, 46, 47, 48, 49, 50, 51, 52, 53, 54, 55, 56, 57, 58, 59,
   60, 61, 62, 63, 64, 65, 66, 67, 68, 69, 70, 71, 72, 73, 74,
   75, 76, 77, 78, 79, 80, 81, 82, 83, 84, 85, 86, 87, 88, 89,
   90, 91, 92, 93, 94, 95, 96, 97, 98, 99 ]
 setting:
   values: [
     "Accumulate",
     "Accumulate-Subsample",
     "Replace",
   ]
 sigma_squared:
   values: [
     0.1, 1.0, 10.
   ]
```

Seeds were swept from 0 to 99, inclusive. Note: We ran this sweep as 9 separate sweeps; to understand why, see this GitHub issue.

# E. Additional Experimental Results for Model Collapse Hyperparameters

Due to space limitations in the main text, we can oftentimes only present a subset of runs corresponding to a subset of hyperparameters. We present additional figures with a wide range of hyperparameters here for completeness.

## E.1. Additional Results for Model Collapse in Linear Regression

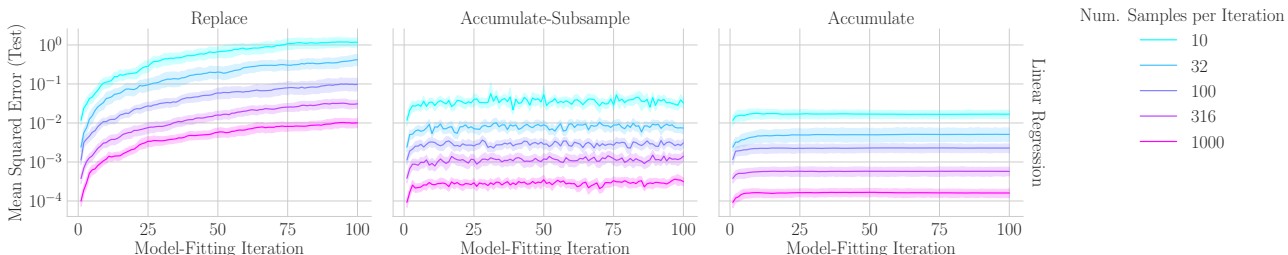

*Figure 6.* Linear Regression for Data Dimension $d = 1$ and variance $\sigma^2 = 0.10$.

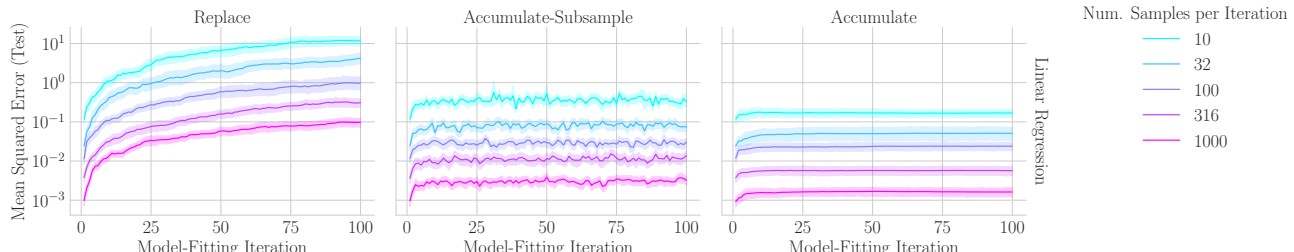

*Figure 7.* Linear Regression for Data Dimension $d = 1$ and variance $\sigma^2 = 1.00$.

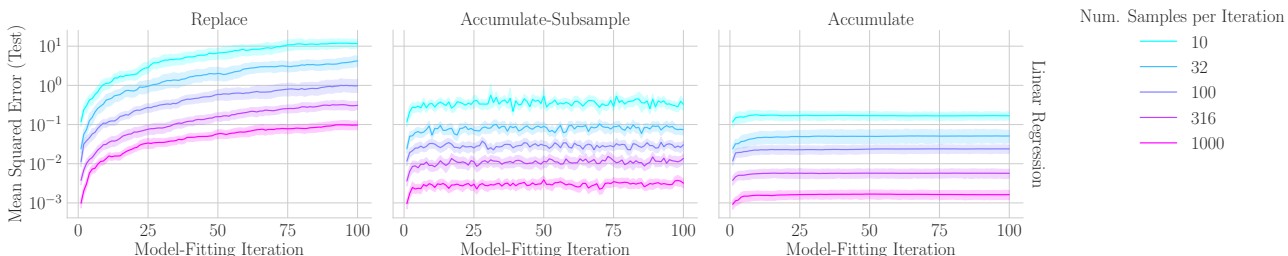

*Figure 8.* Linear Regression for Data Dimension $d = 1$ and variance $\sigma^2 = 10.0$.

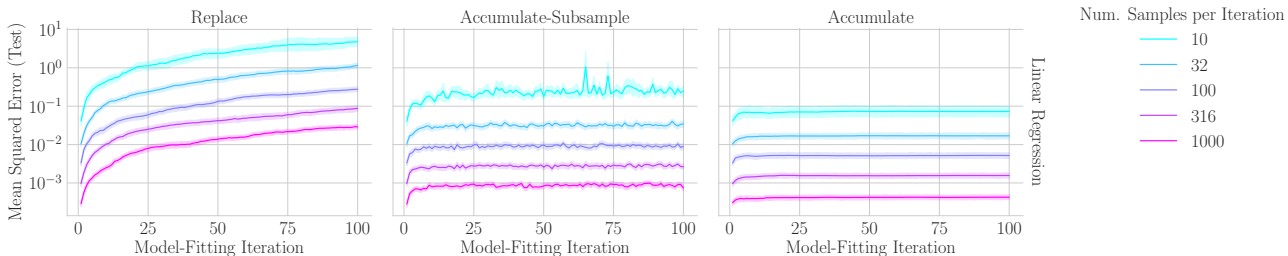

*Figure 9.* Linear Regression for Data Dimension $d = 3$ and variance $\sigma^2 = 0.10$.

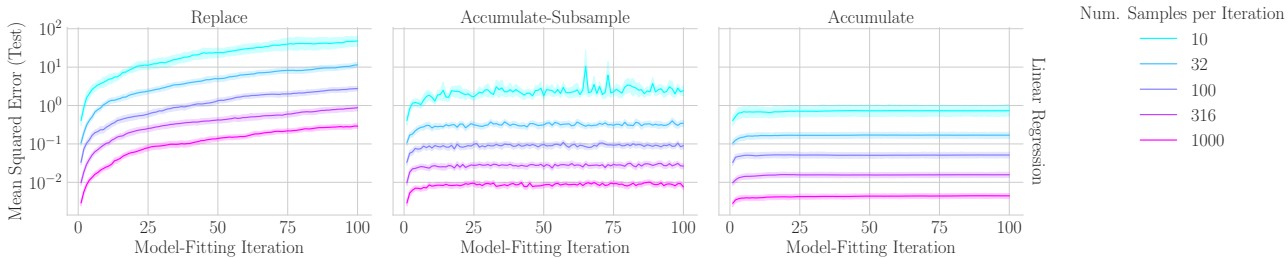

*Figure 10.* Linear Regression for Data Dimension $d = 3$ and variance $\sigma^2 = 1.00$.

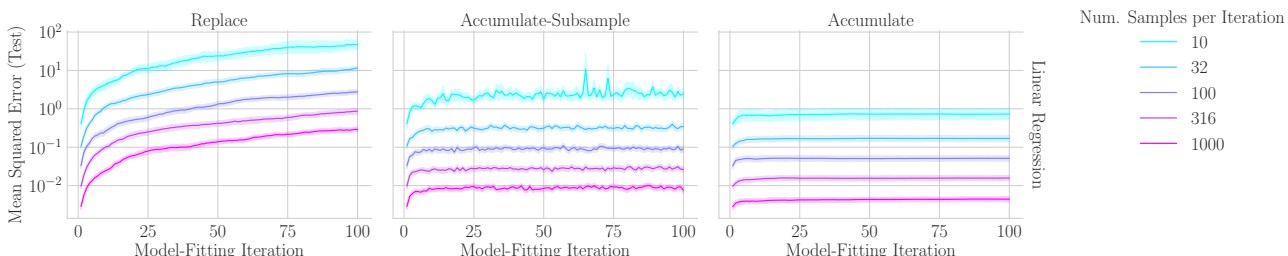

*Figure 11.* Linear Regression for Data Dimension $d = 3$ and variance $\sigma^2 = 10.0$.

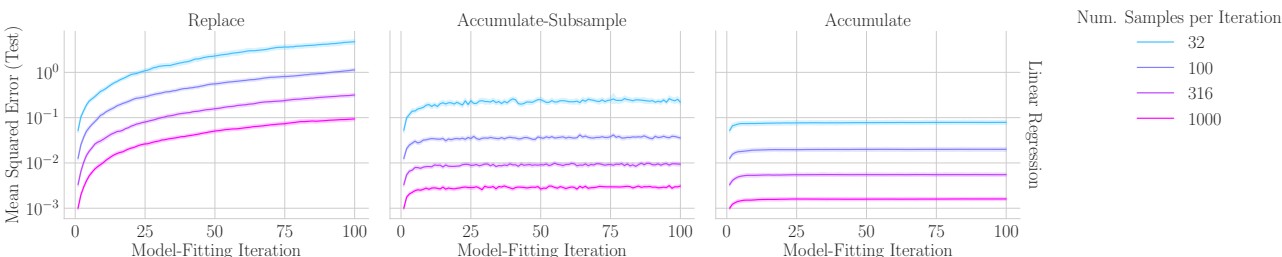

*Figure 12.* Linear Regression for Data Dimension $d = 10$ and variance $\sigma^2 = 0.10$.

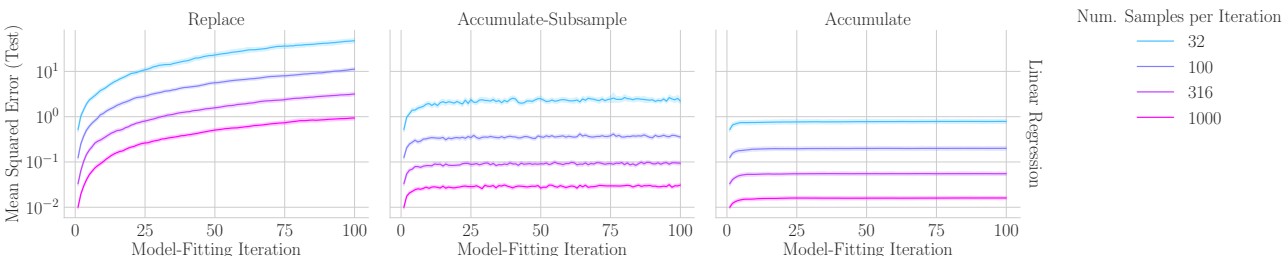

*Figure 13.* Linear Regression for Data Dimension $d = 10$ and variance $\sigma^2 = 1.00$.

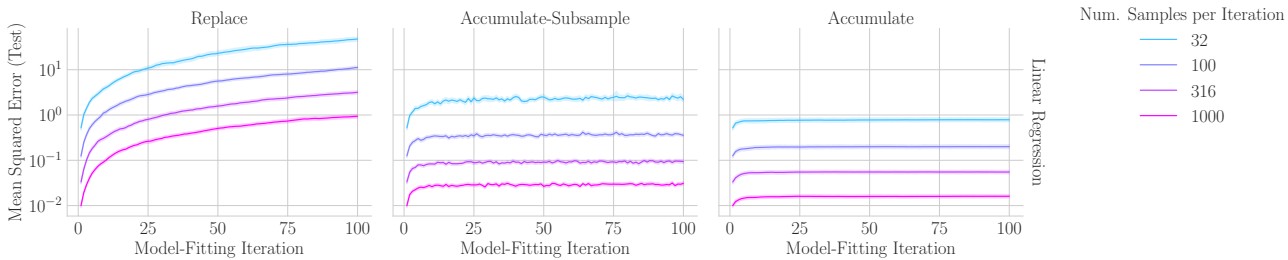

*Figure 14.* Linear Regression for Data Dimension $d = 10$ and variance $\sigma^2 = 10.0$.

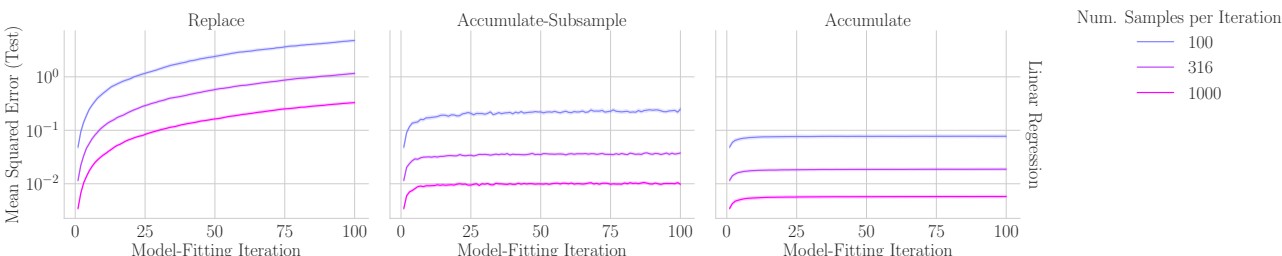

*Figure 15.* Linear Regression for Data Dimension $d = 32$ and variance $\sigma^2 = 0.10$.

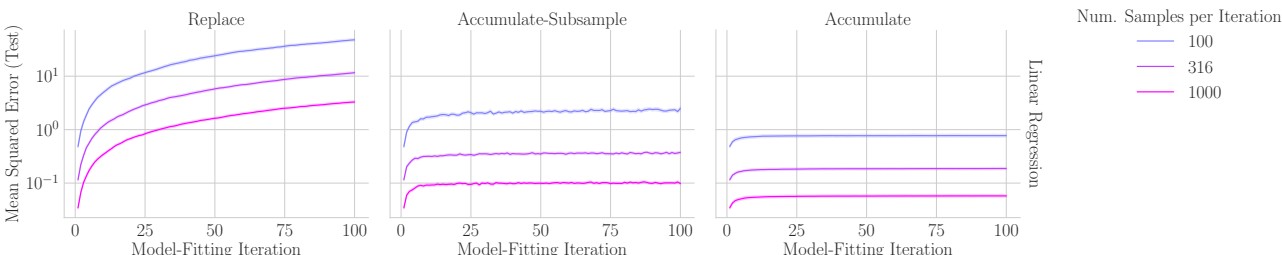

*Figure 16.* Linear Regression for Data Dimension $d = 32$ and variance $\sigma^2 = 1.00$.

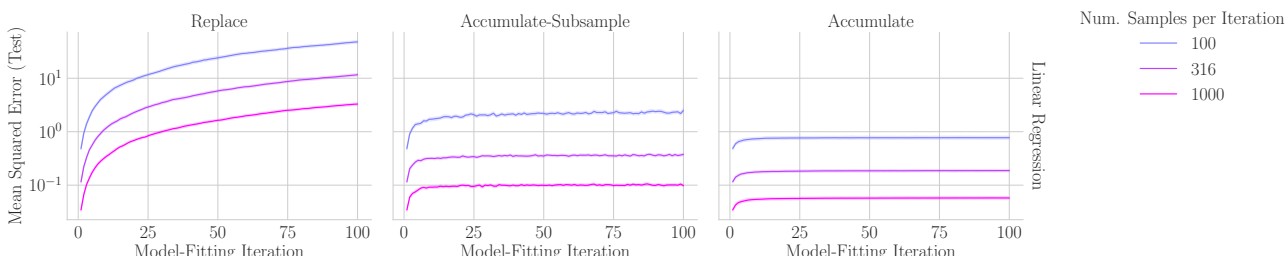

*Figure 17.* Linear Regression for Data Dimension $d = 32$ and variance $\sigma^2 = 10.0$.

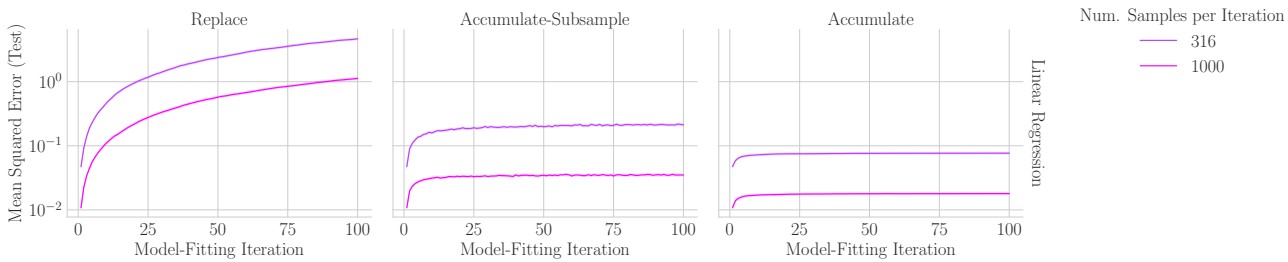

*Figure 18.* Linear Regression for Data Dimension $d = 100$ and variance $\sigma^2 = 0.10$.

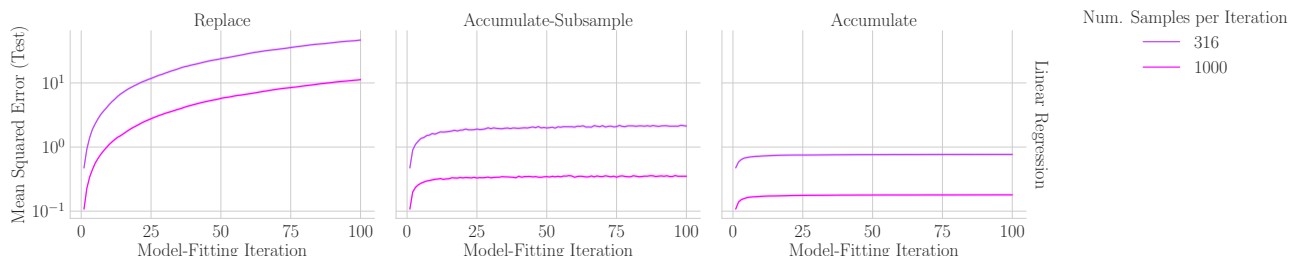

*Figure 19.* Linear Regression for Data Dimension $d = 100$ and variance $\sigma^2 = 1.00$.

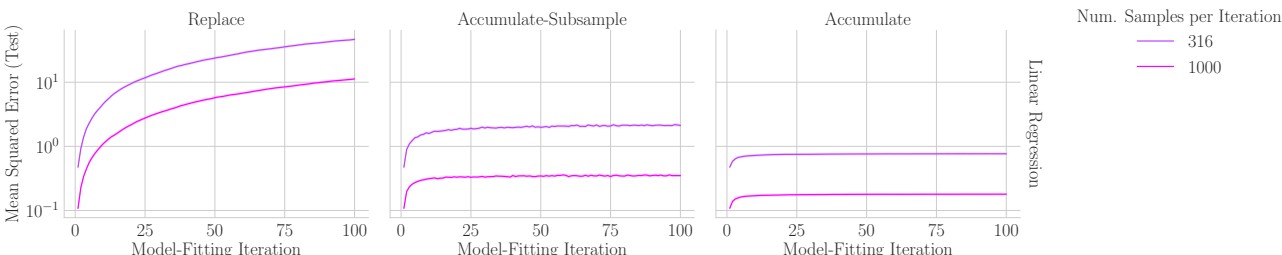

*Figure 20.* Linear Regression for Data Dimension $d = 100$ and variance $\sigma^2 = 10.0$.

