# OpenReview forum: "Collapse or Thrive: Perils and Promises of Synthetic Data in a Self-Generating World"
_ICML.cc/2025/Conference — ICML 2025 poster_

### Official Review · Reviewer_APMk · 2025-02-18

**Overall Recommendation:** 3

**Summary:**

This paper investigates model collapse under different data regimes: 1) when we train the model on the latest synthetic data (the setting used in Shumailov et al.) 2) when we accumulate the synthetic data at each iteration (keeping the original data in the first iteration) 3) when we accumulate and subsample to obtain a fixed dataset size at each iteration. The authors show that in 2) the test losses do not diverge, and in 3), the test losses do not diverge or the divergence is slow.
Another contribution is the investigation of the importance of the size of the real data vs its proportion in the training set.


Update after rebuttal:
I keep my rating of 3.

**Claims And Evidence:**

All the claims are validated with experiments, and some of them with theory.

The empirical evidence encompasses three data settings: 1) multivariate gaussian modeling 2) kernel density estimation 3) SFT of LLMs

The theroretical evidence encompasses the univariate gaussian modeling setting (accumulate workflow), KDE (replace and accumulate workflow).  However, no result is given for the accumulate + subsample setting.

**Essential References Not Discussed:**

Most of the works related to this paper are cited, but the related work section in the appendix could be improved to make the objective and distinctness of each work more clear, namely the experimental setup considered (how is the synthetic data mixed with real data) and the theoretical results.

**Experimental Designs Or Analyses:**

Yes, no issue.

**Methods And Evaluation Criteria:**

The datasets choice mimicks the one adopted in Shumailov et al.
The gaussian modeling and KDE experiments feel more like sanity checks (for which some theoretical results can be proven) than realistic settings where model collapse might appear.

**Other Comments Or Suggestions:**

NA

**Other Strengths And Weaknesses:**

The paper is well-written and easy to follow.

**Questions For Authors:**

Do the authors have some intuition on possible preliminary theoretical results for the accumulate-subsample workflow, even in a very simplified setup?

Could the authors be more clear on how the theoretical results in Dohmatob et al. differ from the insights shown (experimentally) in Section 4?

Can the authors provide results for a more realistic setting than the one used the gaussian and kde experiments. The SFT one conducted in the paper is a good example of realistic scenario.

**Relation To Broader Scientific Literature:**

In the first part of the paper, the objective of the analysis is to extend some insights which were given in Gerstgrasser et al. ([1]) (namely that the accumulate workflow might not lead to divergence) to the experimental setup used in Shumailov et al. ([2]).
Section 4 looks at the importance of the cardinality of the real data vs its proportion. This section is empirical, but previous theoretical results in [3] shared common insights on the potential benefits of synthetic data up to a certain point, when real data is scarce.
Another important reference is [4] which also studies model training on real and synthetic data (with the slight difference that there is no accumulation in [4], as the synthetic data from the latest iteration is mixed with the real data).


[1] Gerstgrasser, M., Schaeffer, R., Dey, A., Rafailov, R., Korbak, T., Sleight, H., Agrawal, R., Hughes, J., Pai, D. B., Gromov, A., Roberts, D., Yang, D., Donoho, D. L., and Koyejo, S. Is model collapse inevitable? breaking the curse of recursion by accumulating real and synthetic data, 2024.

[2] Shumailov, I., Shumaylov, Z., Zhao, Y., Papernot, N., Anderson, R., and Gal, Y. Ai models collapse when trained on recursively generated data. Nature.

[3] Dohmatob, E., Feng, Y., Yang, P., Charton, F., and Kempe, J. A tale of tails: Model collapse as a change of scaling laws. In Forty-first International Conference on Machine Learning, 2024.

[4] Bertrand, Q., Bose, A. J., Duplessis, A., Jiralerspong, M.,and Gidel, G. On the stability of iterative retraining of generative models on their own data. The Twelfth International Conference on Learning Representations, 2024.

**Theoretical Claims:**

Yes, no issue.

---

> ### Author Rebuttal · Authors · 2025-03-30
>
> Thank you for your thoughtful review and recommendation. We appreciate your recognition that our paper is well-written and that our claims are properly validated.
>
> ### Theoretical Results for Accumulate-Subsample
>
> You raise an excellent question. The accumulate-subsample workflow presents significant analytical challenges because of subsampling the data at each model-fitting iteration.  We welcome any suggestions you might have for tackling this theoretical challenge, as we believe it represents an important open problem for the field.
>
> ### Clarification of Relationship to Dohmatob et al. in Section 4
>
> > Could the authors be more clear on how the theoretical results in Dohmatob et al. differ from the insights shown (experimentally) in Section 4?
>
> Dohmatob et al. 2024 show in Corollary 3.3 that in a simplified mathematical setting, the test error on real data scales as $(T_{real} + T_{AI})^{-(1 - 1/\beta)} + k^{-(\beta -1)}$. This means that in this setting, AI-generated data can help improve performance.  Directly comparing our empirical results to their theoretical results is difficult because our generative models differ. Additionally, their result decouples the amount of data $T_{real} , T_{AI}$ from the tail cutoff $k$, whereas decoupling these two factors in real data would be difficult (and potentially not possible). We see our results as complementary, but different, and want to give appropriate credit to prior work. We will clarify these distinctions in the revised paper.
>
> ### Realistic Settings
>
> > Can the authors provide results for a more realistic setting than the one used in the gaussian and kde experiments. The SFT one conducted in the paper is a good example of realistic scenario.
>
> We appreciate this feedback. To explain why we used these classical statistical models, at least three previous papers have studied multivariate Gaussians, making this setting crucial for consistent comparison against their results. These settings offer analytical tractability for theoretical results while still demonstrating the key phenomena.
>
> We’d also like to highlight that our paper contains results from pretraining sequences of language models on real text (TinyStories), as shown in Figure 4.
>
> Could you please clarify what "realistic" properties you're looking for that our current experiments don't address?  Is it a matter of the particular generative model? If so, what model and data combination would you like us to run experiments for?
>
> We'd be happy to expand the language model experiments with additional details in the revision or to conduct new experiments if you have specific suggestions.

---

> > ### Comment · Reviewer_APMk · 2025-04-04
> >
> > Thank you for your rebuttals and for your clarification regarding Dohmatob et al.
> >
> > Regarding the more realistic settings, I was thinking about a tabular setting (for example UCI datasets), and some commonly used model (eg. TabDDPM, CTGAN, TVAE,...)

---

> > > ### Author Response · Authors · 2025-04-06
> > >
> > > Thanks for the suggestion!  We investigated the code base and found that TabDDPM’s code base is currently broken (19 open issues) and we were unable to fix it on short notice to run this experiment.
> > >
> > > Though we appreciate your suggestion, we would like to point out several things:
> > > - We include two "realistic” settings already (pre-training and SFT), which have recently generated great excitement surrounding the use of synthetic data [1,2].
> > > - Pre-training is run on 4 different models across 3 different collapse paradigms.
> > > - The SFT experiments also include 3 models with 3 seeds across 3 different model collapse paradigms.
> > > - The paper also contains 3 sets of statistical experiments (again with 3 settings and 3 seeds each) replete with proofs of asymptotic behavior.
> > >
> > > We do not know of any other model collapse papers on language models that run experiments that are this extensive.  We apologize for being unable to run TabDDPM experiments on short notice, but we hope you’ll understand that our paper already excels when it comes to the breadth and thoroughness of its experiments, both ``realistic” and theoretical.
> > >
> > > [1] https://arxiv.org/abs/2409.07431
> > > [2] https://arxiv.org/pdf/2501.19393

---

### Official Review · Reviewer_uuKC · 2025-03-04

**Overall Recommendation:** 3

**Summary:**

This paper examines the phenomenon of model collapse in generative learning, where models are trained on data that includes synthetic generations from previous iterations. The authors investigate three training paradigms - Replace, Accumulate, and Accumulate-Subsample - and find that Replace leads to model collapse, Accumulate prevents it, and Accumulate-Subsample slows degradation but does not fully eliminate it. The paper provides empirical evidence across multiple generative model settings including multivariate Gaussian estimation, kernel estimation and language model fine tuning.

**Claims And Evidence:**

main claims:
- Replace leads to model collapse, where test loss diverges across iterations. This is demonstrated in multivariate Gaussian modeling, kernel density estimation (KDE), and language model fine-tuning experiments.
- Accumulate prevents collapse, as models trained on both real and synthetic data remain stable.
- Accumulate-Subsample reduces but does not eliminate degradation, acting as a middle ground between Replace and Accumulate.

Issues:
- The paper presents accumulate-subsample as a new paradigm, but prior work (e.g., Dohmatob et al. 2024, Bertrand et al. 2024) already studied real-vs-synthetic data trade-offs in iterative training.
- The paper claims that synthetic data can improve performance in some settings, but this was already explored in Dohmatob et al. (2024) and Seddik et al. (2024).
- Definition of Model Collapse: The paper presents collapse as "test loss divergence", ignoring alternative definitions such as scaling law deviations (Dohmatob 2024), loss of rare modes (Shumailov 2024), and knowledge degradation (Peterson 2024).

**Essential References Not Discussed:**

The paper does discuss and cite many of the relavent work but I believe it does not do just in citing the previous findings.

**Experimental Designs Or Analyses:**

The main issues I see with the experiments is that:
- The paper does not analyze synthetic data quality or filtering, which are key considerations in real-world. For example most text that enters the internet does have some human evaluation on top of it. So I think all the setups in the paper are to some extend unrealistic.

- There is no long-term stability analysis, making it unclear how performance evolves after many more iterations.

**Methods And Evaluation Criteria:**

Although the experimental setup is well-structured, with multiple model classes tested across generative settings, but the experiments focus on data quantity (real vs. synthetic ratios) but ignore data quality. This is an important step which is ignored.

Evaluation criteria are relevant but incomplete, as long-term performance degradation is not analyzed rigorously beyond several iterations specifically for LLMs experiments.

**Other Comments Or Suggestions:**

See below.

**Other Strengths And Weaknesses:**

- The main contributions lack novelty relative to prior work.
- The definition of model collapse is too narrow.
- The paper does not address synthetic data filtering or quality.

**Questions For Authors:**

1 - How does this work differ from Dohmatob et al. (2024), which already studied real-vs-synthetic data trade-offs?

2 - Why does the paper not discuss alternative definitions of model collapse (e.g., rare mode loss, scaling law deviations)?

3 - Can the authors justify why accumulate-subsample prevents collapse beyond empirical results?

**Relation To Broader Scientific Literature:**

- The paper overstates its originality and does not fully acknowledge prior research on model collapse.
- A more comprehensive literature review is needed to position the contributions in relation to existing studies.

**Theoretical Claims:**

Theoretical results are only for Gaussian and KDE models. No theoretical proof for why Accumulate-Subsample prevents full collapse.
And prior work by Bertrand et al. (2024) already provided mathematical stability conditions for iterative training on synthetic data.

---

> ### Author Rebuttal · Authors · 2025-03-30
>
> ### Definitions of Model Collapse
>
> You raise an excellent point. A recent review (https://arxiv.org/abs/2503.03150) identifies multiple definitions of model collapse. We focused on test loss divergence as it addresses an existential threat: all future generative models becoming useless. We'll expand our discussion to acknowledge alternative definitions and clarify our specific focus.
>
> ### Novelty in Accumulate-Subsample Paradigm
>
> We appreciate the opportunity to clarify what distinguishes our work from prior research:
>
> 1. Practical Framing: We specifically examine accumulate-subsample through fixed compute per model-fitting iteration, which directly models real-world ML development constraints. This framing is absent from the papers cited but is absolutely critical. One can study all manner of different workflows for data, but their importance depends on how closely they model the real world.
>
> 2. Training Methodology: Bertrand et al. (2024)  studies _iterative retraining_, where each model is initialized from the previous model and each optimizer is initialized from the previous model’s optimizer state. The iterative retraining approach in Bertrand et al. mitigates model collapse because models are constrained from wandering too far from their predecessors. However, this doesn't reflect how frontier models are developed in practice. In comparison, we model the scenario where successive model generations are trained independently.
>
> 3. Data Evolution: We consider synthetic data accumulation across multiple model generations, whereas prior work typically only considers synthetic data from the most recently trained model e.g., Bertrand & Alemohammad.
>
> We will improve our manuscript to clarify.
>
> ### Related Work
>
> > How does this work differ from Dohmatob et al. (2024), which already studied real-vs-synthetic data trade-offs?
>
> Our work differs substantially from both Dohmatob papers. One examines linear regression specifically, while the other investigates scaling laws. Both are largely theoretical. In contrast, our work is more empirical and spans multiple generative paradigms (Gaussian modeling, KDE, language model fine-tuning) and examines different data treatment workflows across these diverse settings.
>
> > Please move the related work section to the main text.
>
> If accepted, we will move Related Work into the main text after the Introduction. We placed it in the Appendix due to the 8-page limit.
>
> We will expand our related work to include additional relevant papers discovered during the review process e.g., Gillman et al. 2024 and better clarify the positioning of our contributions.
>
> ### Understanding Accumulate-Subsample
>
> > Can the authors justify why accumulate-subsample prevents collapse beyond empirical results?
>
> The accumulate-subsample setting is difficult to study analytically and we were not able to prove satisfactory results. Intuitively, accumulate-subsample slows collapse because each new training batch contains a fraction of real data that provides an "anchor" to the true distribution.
>
> ### Benefits of Synthetic Data
>
> Our contribution isn't showing that synthetic data can help performance - we agree this possibility is well-known across many papers. Rather, one contribution is that such a result holds for kernel density estimation - a  simple yet fundamental statistical model - without sophisticated data manipulations. Another contribution is assessing the role of cardinality or proportionality of synthetic data , which to our knowledge was previously unexplored.
>
> Upon careful review of Seddik et al. (2024), we couldn't find where they demonstrate synthetic data improves performance.
>
> ### Realistic Data Quality
>
> Our experimental design deliberately aimed to model how synthetic data proliferates on the internet in practice:
>
> - Content on the internet is not systematically filtered. Vast amounts of web data is garbage, spam, AI-generated slop, etc.
> - By studying the "pessimistic" scenario without filtering, we establish a baseline understanding of collapse dynamics. Filtering can only improve the quality of models.
> - That said, we agree data quality is an important factor, and we view our work as complementary to research on data quality research. Indeed, we write "Our experiments take a pessimistic viewpoint, in the sense that our experiments pay no attention to the quality of data, whereas in practice, engineers heavily filter data based on various indicators of data quality, e.g., (Brown et al., 2020; Lee et al., 2023; Wettig et al., 2024; Penedo et al., 2024; Li et al., 2024b; Sachdeva et al., 2024); for a recent review, see Albalak et al. (2024)."
>
> > There is no long-term stability analysis, making it unclear how performance evolves after many more iterations.
>
> We ran most experiments for 100 iterations across most settings. Running language modeling pretraining for longer proved beyond our limited compute budget. We're open to extending our experiments.

---

### Official Review · Reviewer_Y9pa · 2025-03-11

**Overall Recommendation:** 2

**Summary:**

The paper investigates model collapse (a phenomenon observed when generative models are trained with output generated from such models) in 3 different settings (replace, accumulate, and accumulate-subsample). They find that model collapse occurs in the replace and the accumulate-subsample setting (but slower). They do not observe model collapse in the last setting (accumulate). Additionally, they investigate how the portion of real and synthetic data in a dataset influences model performance in a supervised fine-tuning regime.

## Update after rebuttal: ##

I would like to thank the authors for their response. However, my concerns about novelty have not been lifted and some of my questions have not been addressed. Therefore, I will keep my initial score.

**Claims And Evidence:**

The claims of the paper (see summary) are supported by a wide range of experiments and in some cases theoretical analyses. The experiments are convincing and support the claims. I do not agree with the authors that model collapse is avoided in the accumulate-subsample case (judging from their experimental data) but rather slowed down.

**Essential References Not Discussed:**

I suggest that the authors discuss other work more that investigates mixing real and generated data in the model collapse setting (see above).

**Experimental Designs Or Analyses:**

The experimental design seems to be extensive and convincing.

**Methods And Evaluation Criteria:**

The authors evaluate their claims for multivariate gaussian models, kernel density estimation, and supervised fine-tuning of language models for the replace and accumulate setting. They also provide theoretical insights in those cases. For the accumulate-subsample setting, they provide experiments for the 3 aforementioned cases as well as linear regression and pretraining language models. For the last part (section 4) experiments for supervised fine-tuning are presented.

**Other Comments Or Suggestions:**

References:

- Alemohammad, S., Casco-Rodriguez, J., Luzi, L., Humayun, A. I., Babaei, H., LeJeune, D., ... & Baraniuk, R. Self-Consuming Generative Models Go MAD. In The Twelfth International Conference on Learning Representations. (2024)
- Bertrand, Q., Bose, J., Duplessis, A., Jiralerspong, M., & Gidel, G. On the Stability of Iterative Retraining of Generative Models on their own Data. In The Twelfth International Conference on Learning Representations. (2024)
- Briesch, M., Sobania, D., & Rothlauf, F. (2023). Large language models suffer from their own output: An analysis of the self-consuming training loop. arXiv preprint arXiv:2311.16822.
- Dohmatob, E., Feng, Y., Yang, P., Charton, F., & Kempe, J. (2024, July). A tale of tails: model collapse as a change of scaling laws. In Proceedings of the 41st International Conference on Machine Learning (pp. 11165-11197).
- Dohmatob, E., Feng, Y., Subramonian, A., & Kempe, J. (2024). Strong model collapse. arXiv preprint arXiv:2410.04840.
- Gerstgrasser, M., Schaeffer, R., Dey, A., Rafailov, R., Sleight, H., Hughes, J., ... & Koyejo, S. (2024). Is model collapse inevitable? breaking the curse of recursion by accumulating real and synthetic data. arXiv preprint arXiv:2404.01413.
- Gonzalo Mart´ınez, Lauren Watson, Pedro Reviriego, Jose Alberto Hern ´ andez, Marc Juarez, and Rik ´ Sarkar. Towards understanding the interplay of generative artificial intelligence and the internet. arXiv preprint arXiv:2306.06130, 2023.

**Other Strengths And Weaknesses:**

While I found the results and experiments in section 4 interesting, it seems to me that this section is another topic than the main paper. It felt out of place reading this section after the paper so far. To me section 4 is more about how to mix data in a dataset but leaves out the context of model collapse.

**Questions For Authors:**

-	Q1: The paper focuses heavily on test loss in their experiments. Did the authors also consider other metrics and evaluations that are important for generative models  (e.g. diversity biases as mentioned in Martinez et al., Briesch et al., and Guo et al. or)?

**Relation To Broader Scientific Literature:**

- My biggest concern with the paper is its novelty. It builds heavily on Gerstgrasser et al. and section 2 (a large part of the paper) only provides new results for different kinds of generative models with equal conclusions. The second part of the paper (accumulate-subsample) provides empirical insights that mixing real and generated data slows down model collapse. This has been observed and discussed in literature as early as Bertrand et al., Alemohammad et al. and even for the case of language models in Briesch et al. The last section 4 is also close to the results of Dohmatob et al. A (as the authors note correctly).
- I also want to note that the literature is not unanimous on the claim that accumulating data prevents model collapse. That is an ongoing discussion. (e.g. Dohmatob et al. B).

**Theoretical Claims:**

Theoretical results are provided for the results in section 2. I did not check the detailed proof in the appendix for correctness.

---

> ### Author Rebuttal · Authors · 2025-03-30
>
> ### On Novelty
>
> > My biggest concern with the paper is its novelty. It builds heavily on Gerstgrasser et al. and section 2 (a large part of the paper) only provides new results for different kinds of generative models with equal conclusions.
>
> We disagree with this characterization. Our Section 2 identifies and resolves a fundamental tension in the literature where competing papers have reached contradictory conclusions about model collapse. This systematic evaluation across multiple generative settings provides valuable scientific evidence about the generality of these phenomena.
>
> Moreover, we make several fundamental contributions:
> 1. We discovered that in KDE settings (Figure 2), certain combinations of real and synthetic data actually outperform models trained on real data alone - a counterintuitive result not previously reported for KDEs. KDEs are appealing because they are a widely used statistical model that are straightforward to experiment with and analytically tractable to study.
> 2.  We provide novel theoretical results (Theorem 1) on the limiting distribution for univariate Gaussians under the accumulate training workflow, proving that both the covariance and mean errors converge to non-zero constants - in stark contrast to the replace setting where they diverge. This theoretical result mathematically characterizes why the accumulate workflow avoids collapse.
> 3.  Our work is the first to systematically compare all three workflows (replace, accumulate, accumulate-subsample) across five different generative settings, providing a more comprehensive understanding of when and how model collapse occurs.
>
> ### On Accumulate-Subsample and Prior Work
>
> > The second part of the paper (accumulate-subsample) provides empirical insights that mixing real and generated data slows down model collapse. This has been observed and discussed in literature as early as Bertrand et al., Alemohammad et al. and even for the case of language models in Briesch et al.
>
> We should do a much better job explaining how our approach contributes to the literature. While others do study mixing real and generated data, our work pushes the field forward in important ways:
>
> Reason #1: We frame our analysis from an extremely practical real-world engineering constraint: finite compute. Other papers study model-data feedback loops in ways that aren't clearly motivated by practical considerations. This framing is important.
>
> Reason #2: Other papers add complications that make drawing clear insights difficult. For instance, Bertrand et al. 2023 studies _iterative retraining_, where each new model is initialized from the previous model's parameters and the each new model’s optimizer is initialized from the previous model's optimizer’s state This helps mitigate model collapse because each model is then limited in how far it moves from the preceding model, but this iterative retraining poorly matches real-world conditions as frontier models aren't (to the best of our knowledge) initialized from their predecessors' parameters.
>
> Reason #3: The real+synthetic data loops of previous papers e.g., Bertrand and Alemohammad both focus on real data plus synthetic data drawn only from the most recent model. But in reality, if one is scraping the web, we know of no way to filter the collected data for data generated by the most recent generation of models. Our Accumulate-Subsample setting is meant to more faithfully model reality: synthetic data are added online, and are vacuumed up with real data to train the next model.
>
> Reason #4: Briesch et al. 2023 is difficult to interpret because the results of their v1 and v2 Arxiv manuscripts (specifically, the key Figure 4) appear to reach slightly contradictory conclusions that the authors don't address.
>
> In the revised manuscript, we will better contextualize our contributions relative to these prior works, explicitly highlight the practical relevance of our approach, and more clearly explain how our findings extend beyond what was previously known.
>
> ### Definitions and Realistic Assumptions
>
> > I also want to note that the literature is not unanimous on the claim that accumulating data prevents model collapse. That is an ongoing discussion. (e.g. Dohmatob et al. B).
>
> We agree that the field doesn't have a consensus. This is partially because the field has defined model collapse in multiple and sometimes conflicting ways (a point made by a recent position paper https://arxiv.org/abs/2503.03150) and partially because different papers make assumptions that are unrealistic.
>
> This is why we feel it is so important to construct experiments in a way that emulates reality as faithfully as possible. In our opinion, the field has drawn conclusions from experimental setups that are oftentimes not well grounded in real-world considerations.

---

### Official Review · Reviewer_NdVu · 2025-03-20

**Overall Recommendation:** 1

**Summary:**

The manuscript studies three ways of using synthetic data, both empirically and theoretically.

The manuscript starts by examining Gerstgrasser et al. 2024's two claims with the proposed three generative modeling settings in section 2, where some well-established settings and tasks are directly used, making the exact contribution of sec 2 unclear.

Section 3 considers a slightly different setting, namely training with a fixed compute budget, and then studies the value of synthetic data for reducing test loss on real data depending on the amount of real data.

**Claims And Evidence:**

It is hard to distinguish between the authors' contributions and the existing papers. E.g.,
* The manuscript argues that "Following Shumailov et al. (2023); Alemohammad et al. (2024); Bertrand et al. (2024), we study what happens when one iteratively fits multivariate Gaussians and samples from ..." and "Shumailov et al. (2024) proved that as t → ∞". It seems like the manuscript just reproduces these experiments in a unified setting with some take messages. Similar comments can be applied to Sec 2.2 and Sec 2.3.
* The evidence in Sec 4 is not very rigorous to answer the key open question mentioned in the first paragraph of Sec 4. E.g.,
    * it is unclear why we can compute the $\log (real / (real + synthetic))$ to best capture the relationship between the fraction of real data and the log-likelihood.
    * it is unclear how to get the value of $1,024$ to support the claim that "when the number of real data is 1024 or lower, we find that there is a small but non-zero amount of synthetic data that improves the test loss when it is included."

**Essential References Not Discussed:**

NA

**Experimental Designs Or Analyses:**

Yes. See comments above.

**Methods And Evaluation Criteria:**

Looks ok. The main concern here is that evaluations and settings in Sec 2 reuse the existing work.

**Other Comments Or Suggestions:**

## update after rebuttal
It would be great if the authors could include more numerical results.
For example, in section 4, fine-tuning different base models on various real datasets with different dataset sizes would be very helpful.
- The training hyper-parameters used in the experiments are unclear. It would be great if the authors could report these and provide a comprehensive ablation study.

**Other Strengths And Weaknesses:**

## update after rebuttal
Strengths:
- The paper is well-structured. It considers problems like Multi-variate Gaussian Modeling, Kernel Density Estimation, Linear Regression, and Language Modeling.

Weaknesses:
- The reviewer acknowledges the additional experiments provided by the manuscript. However, the reviewer does not think it has made sufficient contributions to make it above the bar of ICML acceptance.
- The reviewer believes in adding extensive results on various configurations/settings of recursive training image diffusion models and language autoregressive models. For example, settings include but are not limited to different numbers of training samples in the original real dataset, the different ratios between real and synthetic datasets, and different types of real-world datasets.

**Questions For Authors:**

See comments above.

**Relation To Broader Scientific Literature:**

NA

**Theoretical Claims:**

The reviewer checks the theorem in the main text and does not spot the issues.

---

> ### Author Rebuttal · Authors · 2025-03-29
>
> While we appreciate your review, you are mistaken about your claim that we primarily replicated experiments from other works, and the questions that you asked were largely answered in the paper. To explain line-by-line.
>
> > It seems like the manuscript just reproduces these experiments [from Bertrand, Alemohammad, and Shumailov] in a unified setting with some take messages.
>
> This is false.
>
> Shumailov: Shumailov proves that model collapse occurs for KDEs and Gaussian fitting in the REPLACE SETTING.  A main point of our paper is that if you repeat these experiments IN THE ACCUMULATE SETTING, then collapse does not occur.  We are the first to do this, and thereby directly compare and contrast the replace and accumulate settings.
>
> Alemohammad: This paper again tests model collapse for generative image models in the replace setting.  We don’t test generative image models at all in this paper, and we don’t use the replace setting.
>
> Bertrand: Bertrand tests model collapse for language models in a variation of the accumulate setting.  In this case, the proportion of real data is fixed but non-zero throughout all iterations.  We allow the fraction of real data to go to 0 in our version of the accumulate setting, which marks a significant conceptual difference from Bertrand’s setup.
>
> **All experiments in this paper are original, except for the pretraining experiment, which we extend from Gerstgrasser et. al. to include more model-fitting iterations.**
>
> Overall, our Section 2 makes a novel and important contribution by identifying seemingly contradictory claims in the model collapse literature and comparing them in a head-to-head manner to resolve them.
>
> >The evidence in Sec 4 is not very rigorous to answer the key open question mentioned in the first paragraph of Sec 4.
>
> We provide an F-test of whether the proportion or the cardinality of real data dictates model performance.  This is a standard and fully rigorous statistical evaluation. The high statistical significance (p-values of $6.9e-25$ and $4.6e-25$ strongly supports our conclusions. If you disagree, please let us know why and what you would recommend instead.
>
> > It is unclear how to get the value of $1,024$ to support the claim that "when the number of real data is 1024 or lower, we find that there is a small but non-zero amount of synthetic data that improves the test loss when it is included."
>
> This value is our result, shown in Figure 5, where the curves for real data samples ≤1024 show improvement with synthetic data addition, while curves for samples >1024 do not show this benefit. We are not making a theoretical claim about the number 1024; we are reporting the empirical result we observed across multiple experimental runs.
>
> We believe these clarifications help address the reviewer's concerns and demonstrate both the novelty and rigor of our work. Additionally, the review does not discuss any strengths of the paper, and you left most of your review sections blank.  We hope that you can provide more information about why you gave such a low score and also provide specific constructive criticism for how we can improve the manuscript.

---

### Decision · Program_Chairs · 2025-05-01

**Decision:**

Accept (poster)

**Comment:**

The reviewers appreciated this paper's comprehensive investigation of how different training workflows impact model collapse across multiple generative settings. While some concerns were raised, for example, about novelty relative to prior work, others recognized the paper's empirical contributions and theoretical analysis demonstrating that accumulating both real and synthetic data prevents model collapse, whereas replacing real data with synthetic data leads to divergence. Overall, this paper provides valuable insights through rigorous experiments across diverse settings and offers a useful framework for understanding when synthetic data can be beneficial versus detrimental. The exposition of these findings makes this a worthwhile contribution to the understanding of synthetic data for training.